# Robust membrane protein tweezers reveal the folding speed limit of helical membrane proteins

Seoyoon Kim[1†], Daehyo Lee[1†], WC Bhashini Wijesinghe[1], Duyoung Min[1,2]*

[1]Department of Chemistry, Ulsan National Institute of Science and Technology, Ulsan, Republic of Korea; [2]Center for Wave Energy Materials, Ulsan National Institute of Science and Technology, Ulsan, Republic of Korea

**Abstract** Single-molecule tweezers, such as magnetic tweezers, are powerful tools for probing nm-scale structural changes in single membrane proteins under force. However, the weak molecular tethers used for the membrane protein studies have limited the observation of long-time, repetitive molecular transitions due to force-induced bond breakage. The prolonged observation of numerous transitions is critical in reliable characterizations of structural states, kinetics, and energy barrier properties. Here, we present a robust single-molecule tweezer method that uses dibenzocyclooctyne cycloaddition and traptavidin binding, enabling the estimation of the folding 'speed limit' of helical membrane proteins. This method is >100 times more stable than a conventional linkage system regarding the lifetime, allowing for the survival for ~12 hr at 50 pN and ~1000 pulling cycle experiments. By using this method, we were able to observe numerous structural transitions of a designer single-chained transmembrane homodimer for 9 hr at 12 pN and reveal its folding pathway including the hidden dynamics of helix-coil transitions. We characterized the energy barrier heights and folding times for the transitions using a model-independent deconvolution method and the hidden Markov modeling analysis, respectively. The Kramers rate framework yields a considerably low-speed limit of 21 ms for a helical hairpin formation in lipid bilayers, compared to μs scale for soluble protein folding. This large discrepancy is likely due to the highly viscous nature of lipid membranes, retarding the helix-helix interactions. Our results offer a more valid guideline for relating the kinetics and free energies of membrane protein folding.

*For correspondence: dymin@unist.ac.kr

[†]These authors contributed equally to this work

Competing interest: The authors declare that no competing interests exist.

## Editor's evaluation

This work presents an important advance of single-molecule force spectroscopy of membrane proteins providing a new robust design of the linkage between a target single molecule and solid support. The data provide compelling evidence of the improved mechanical stability of the pulling system for more statistically reliable force measurements of bio-macromolecules. Also, the quantification of speed-limit of membrane protein folding is exquisite and informative, representing an important contribution to the field.

## Introduction

Single-molecule magnetic tweezers have emerged as a powerful tool for studying membrane proteins, providing valuable insights into their biogenesis, interactions with drugs, and the effects of chaperones on their folding (*Hong et al., 2022*; *Petrosyan et al., 2021*; *Jefferson et al., 2018*). Their extremely low spring constants allow for constant low-force measurements (*Wang et al., 2019*). This advantage has made it possible to observe low-energy transmembrane (TM) helix interactions of

helical membrane proteins within lipid bilayers (*Choi et al., 2019*; *Choi et al., 2022*). Rapid noncovalent tethering approach using digoxigenin (dig) and biotin is commonly used to attach target molecules to solid supports (beads or chamber surface; *Choi et al., 2022*; *Bauer et al., 2022*; *Huang et al., 2022*; *Moessmer et al., 2022*; *Pandey et al., 2022*; *Zhao and Woodside, 2021*; *Maciuba et al., 2021*; *Avellaneda et al., 2020*; *Gupta et al., 2020*; *Kostrz et al., 2019*; *Jiao et al., 2018*; *Min et al., 2018*; *Hao et al., 2017*; *Yao et al., 2014*; *del Rio et al., 2009*). However, the bound complexes dissociate at force ranges of tens of pN (*Van Patten et al., 2018*; *Janissen et al., 2014*; *Schlingman et al., 2011*). Especially, the dig-antidig binding has a short lifetime of ~1 s at 25 pN and ~0.1 s at 50 pN (*Van Patten et al., 2018*), limiting the stable performance of experiments. This poses a bottleneck in membrane protein studies using single-molecule tweezers.

Multiple biotins and digs have been used to improve the tethering of DNA molecules (*Kaczmarczyk et al., 2020*; *Berghuis et al., 2016*). However, the tags are sparsely distributed in the DNA handles attached to a target DNA, with ~1 tag per every 14–20 bp. Thus, the detachment of one or more tags during experiments could interfere with small protein (un)folding signals of a few to tens of nm. Thiol- or amine-based covalent tethering approaches *Janissen et al., 2014*; *Schlingman et al., 2011* have limitations in studying proteins because the reactions can also attack native residues like cysteine and lysine, as well as the N-termini of proteins. Protein/peptide tags like HaloTag and SpyTag have also been used to covalently tether tandem repeat constructs to the solid supports (*Alonso-Caballero et al., 2021a*; *Dahal et al., 2020*; *Popa et al., 2016*; *Chen et al., 2015*; *Alonso-Caballero et al., 2021b*). However, the methods can be restricted to specific proteins that are less vulnerable to aggregation. For example, the tandem repeat construct of maltose-binding protein has been used as a protein aggregation model to study the effects of chaperone proteins (*Mashaghi et al., 2016*; *Bechtluft et al., 2007*). Hence, these methods are unlikely to be applicable to membrane proteins, which are highly hydrophobic and prone to the aggregation.

Furthermore, in all covalent protein-tethering approaches, at least one end of a protein or its tandem repeat is attached to a solid support without using long molecular spacers (*Alonso-Caballero et al., 2021a*; *Dahal et al., 2020*; *Popa et al., 2016*; *Chen et al., 2015*; *Alonso-Caballero et al., 2021b*; *Tapia-Rojo et al., 2023*; *Guo et al., 2021*; *Tapia-Rojo et al., 2020*; *Yu et al., 2020*; *Löf et al., 2019*). This close proximity to the surface and/or between proteins of the tandem repeat can hinder membrane protein studies since lipid bilayer mimetics such as bicelles or vesicles of tens of nm or larger in diameter would not be accommodated properly in the narrow space (*Choi et al., 2019*; *Choi et al., 2022*; *Min et al., 2018*; *Min et al., 2015*). On the other hand, single-molecule tweezers using long DNA handles flanking a protein of interest still rely on weak noncovalent tethers, such as the dig-antidig (*Choi et al., 2022*; *Moessmer et al., 2022*; *Pandey et al., 2022*; *Zhao and Woodside, 2021*; *Maciuba et al., 2021*; *Avellaneda et al., 2020*).

Here, we established highly stable membrane protein tweezers, which employ three orthogonal tethering strategies using dibenzocyclooctyne (DBCO), traptavidin, and SpyTag. This tethering system has a lifetime of ~12 hr at 50 pN, a force level sufficient for most biomolecular interactions, covering up to 98% of surveyed protein folds. The system stability is over 100 times higher than the conventional dig/biotin system with a lifetime of only ~7 min at 45 pN (*Janissen et al., 2014*). The tethering strategies were tailored to membrane proteins, which differs from previous DBCO conjugation methods for DNA molecules (*Lin et al., 2023*; *Lin et al., 2021*; *Eeftens et al., 2015*), which will be discussed in Results.

Using our tweezer method with an improved stability, we were able to observe the incessant structural changes of a 4TM-helix bundle protein for a prolonged 9 hr. This long-term observation of numerous transitions enabled us to map a folding energy landscape including the energy barriers ($\Delta G^{\dagger}$), using a deconvolution method independent of kinetic models (*Gebhardt et al., 2010*; *Woodside et al., 2006*). Additionally, we employed the hidden Markov modeling (HMM) analysis to extract the folding times (*Zhang et al., 2016*; *Jiao et al., 2017*) and corrected the values for instrumental errors (*Jacobson and Perkins, 2020*; *Cossio et al., 2015*; *Neupane and Woodside, 2016*; $\tau_{folding}$; the reciprocal of the folding rate constant, $k_{folding}$). In the Kramers rate theory ($\tau_{folding} = \tau_{\omega} \cdot \exp[\Delta G^{\dagger}/k_B T]$; *Hänggi et al., 1990*), the pre-exponential factor ($\tau_{\omega}$) provides an estimate of the fastest possible folding time for a protein (*Kubelka et al., 2004*; *Chung and Eaton, 2018*). Thus, the time scale of $\tau_{\omega}$ is considered as the so-called 'speed limit' of protein folding (*Kubelka et al., 2004*; *Chung and Eaton, 2018*). Using the two obtained parameters and the Kramers equation,

we estimated the speed limits of structural transitions of the helical membrane protein in a lipid bilayer environment. Among three distinct transitions, the formation of a helical hairpin resulted in a significantly reduced speed limit by a factor of ~$10^4$, compared to soluble proteins of similar size (*Kubelka et al., 2004*). Since a helical hairpin is the minimal structural size for helical membrane proteins that require the tertiary folding (*Choi et al., 2019*; *Corin and Bowie, 2022*; *Krainer et al., 2018*; *Engelman and Steitz, 1981*), the estimate could be around the folding speed limit of helical membrane proteins.

## Results

### Assembly of our single-molecule tweezer system

Our tweezer approach, depicted in *Figure 1*, involves a step-by-step procedure for single-molecule tethering, as shown in *Figure 2*. The procedure includes the surface modification of magnetic beads (*Figure 2A*), preparation of DNA handles (*Figure 2B*), passivation of sample chamber surface (*Figure 2C*), and the final assembly (*Figure 2D*). In previous DBCO conjugation approaches for DNA tethering to sample chamber surface (*Lin et al., 2023*; *Lin et al., 2021*; *Eeftens et al., 2015*), the surface was passivated with a short polyethylene glycol (PEG) or longer poly(L-lysine)-grafted PEG to prevent nonspecific adsorption (PEG unit [n]=4 and 45, respectively). However, these surface passivation methods may not be appropriate for membrane protein studies, as stable mechanical manipulations of single membrane proteins have been achieved using a much longer PEG with over 100 units (*Choi et al., 2019*; *Choi et al., 2022*; *Min et al., 2018*; *Min et al., 2015*). Circumventing this issue, we utilized the long PEG polymer (n=114) for the surface passivation and searched for optimal conditions for successful DBCO conjugation on magnetic beads even in detergent/lipid environments necessary for membrane proteins.

To test the method, we adopted two biomolecules with different hydrophobic properties: a designer membrane protein with four TM helices (scTMHC2; *Lu et al., 2018*), which is highly hydrophobic, and a DNA hairpin with a 17 bp stem and a 6T loop (17S6L), which is highly hydrophilic (*Figure 1*). We conjugated scTMHC2 to 1024 bp long DNA handles at its N- and C-terminal ends (*Min et al., 2016*; *Figure 2B* and *Figure 1—figure supplement 1*). The two DNA handles, with a total contour length of ~700 nm, provide ample space to accommodate a bicelle of tens of nm in diameter, which is a lipid bilayer disc wrapped in detergents (*Choi et al., 2019*; *Min et al., 2018*). It is crucial to reconstitute the lipid bilayer environment to maintain the native fold and stability of membrane proteins (*Min et al., 2015*). The DNA handles also mitigate nonspecific interactions between the solid supports, including beads, and the target membrane protein. The possibility of protein aggregation is also eliminated by pulling a single protein with the DNA handles, rather than a tandem repeat of proteins.

We employed the SpyTag/SpyCatcher conjugation for the DNA handle attachment (*Min et al., 2016*; *Zakeri et al., 2012*; *Figure 2B* and *Figure 1—figure supplement 1*). The maleimide-modified DNA handle was first attached to a cysteine that was engineered in SpyCatcher. The SpyCatcher-labeled DNA handle was then attached to SpyTag (*Zakeri et al., 2012*), which is a 13-residue peptide tag located at the protein ends. Once the SpyCatcher and SpyTag are bound, an isopeptide bond between the pair is spontaneously formed in 1–2 hr at 20–22°C. The other end of the handles was modified with either azide or dual biotins (2×biotin), allowing the molecular construct to be tethered to a magnetic bead and the chamber surface, respectively (*Figures 1 and 2*).

Traptavidin with four biotin-binding pockets is attached at the 2×biotin-modified DNA handle (*Figure 2D*). The traptavidin is a streptavidin variant with ~17-fold slower biotin dissociation (off-rate constant ratio: $k_{off,S}/k_{off,T}$ = 16.7) and ~eightfold stronger biotin binding (dissociation constant ratio: $K_{d,S}/K_{d,T}$ = 8.3; *Chivers et al., 2010*), which can easily replace streptavidin variants typically used in single-molecule tweezers. We then tethered the traptavidin-attached construct to the chamber surface, which was passivated with a high concentration of biotinylated PEG (biotin-PEG) among methyl PEG (mPEG; ~1 biotin-PEG per 5×5 mPEG molecules; *Figure 2D*; 10 min at 20–22°C). The high concentration and long polymer arm (~32 nm contour length) of the biotin-PEG (*Ma et al., 2014*) will likely enable the capture of all remaining biotin-binding pockets of the traptavidin. The mPEG polymer cushion prevents nonspecific adhesion of the sticky DBCO-coated beads and the highly hydrophobic membrane protein (*Figure 2C*).

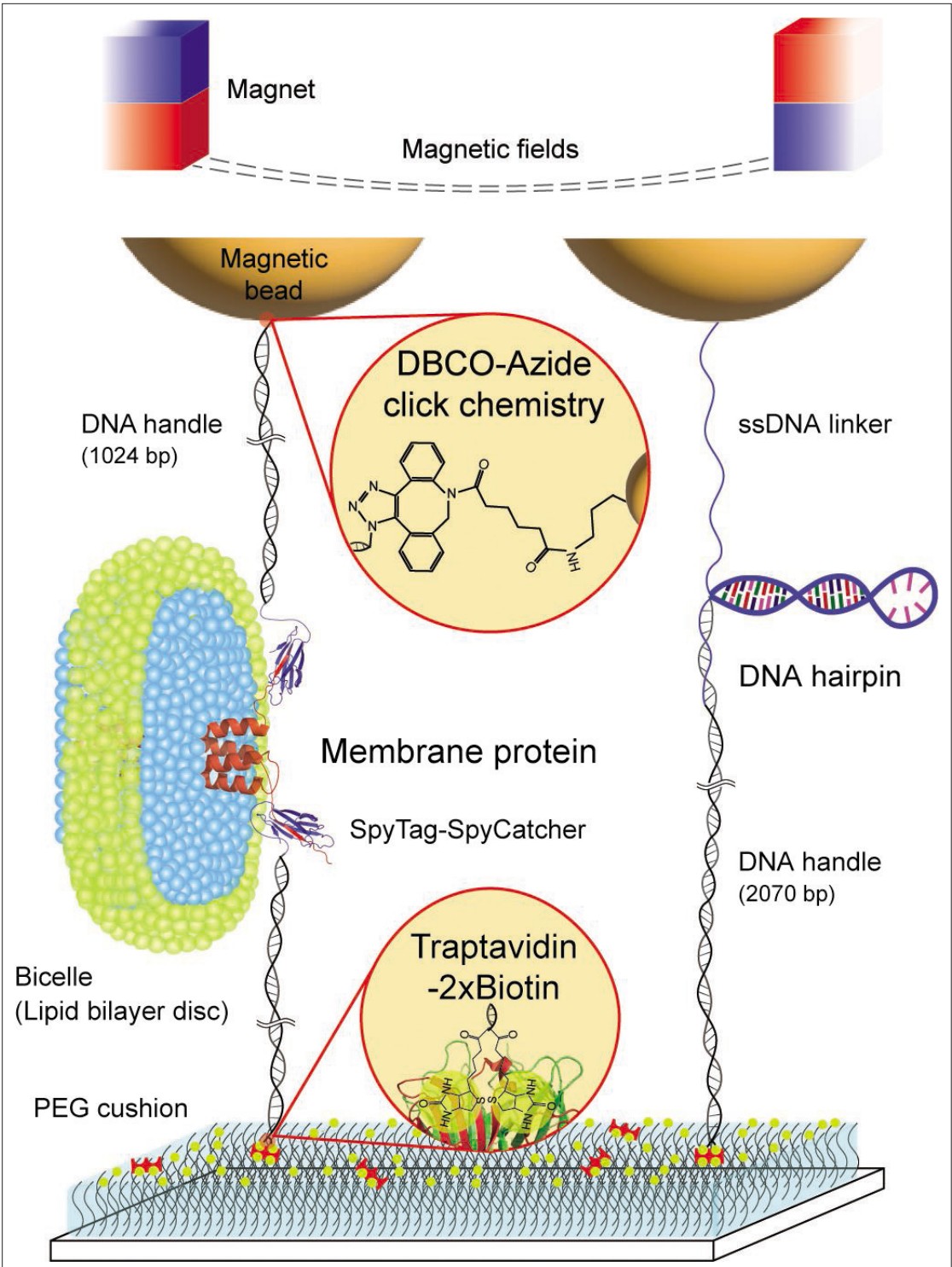

**Figure 1.** Schematic diagram of our robust single-molecule tweezers. The tweezer approach was tested using two biomolecules of a membrane protein, scTMHC2, and a DNA hairpin, 17S6L. DNA handles are attached to the membrane protein via spontaneous isopeptide bond formation between SpyTag and SpyCatcher. A DNA handle is attached to the DNA hairpin via annealing and ligation. One end of the molecular constructs is tethered to magnetic beads via dibenzocyclooctyne (DBCO)-azide conjugation. The other end of the constructs is tethered to the sample chamber surface via traptavidin binding to dual biotins (2×biotin). The surface was passivated by molecular cushion of polyethylene glycol (PEG) polymers. The dashed lines on the top represent magnetic fields generated by a pair of permanent magnets. Lipids and detergents of the bicelle are denoted in blue and green, respectively.

The online version of this article includes the following source data and figure supplement(s) for figure 1:

*Figure 1 continued on next page*

*Figure 1 continued*

**Source data 1.** Original files of raw unedited gels and figures with uncropped gels.

**Figure supplement 1.** Preparation of molecular samples.

## Optimizing conditions for successful single-molecule tethering

We employed the conjugation between the DBCO and azide moieties to attach the other end of the molecular construct to a magnetic bead. In this reaction, the azide-modified end of the construct was conjugated to the DBCO-coated bead without any catalytic enzymes or metal ions (*Figure 2A*). This simple conjugation process provides an advantage over other types of click reactions requiring transition metals or enzyme-catalyzed conjugation approaches (*Kim and Koo, 2019*; *Zhang et al., 2018*). To this end, we labeled DBCO functional groups on amine-modified beads using a DBCO-*N*-hydroxysuccinimide (NHS) ester crosslinker. The DBCO modification on the bead surface was a critical step. Lower or higher concentrations, such as 0.01 or 1 mM, resulted in unsuccessful conjugation to target molecules or nonspecific binding of beads to the surface (*Figure 3* and *Figure 3—source data 1*). We found that using 0.025–0.1 mM DBCO crosslinker was an optimal condition.

It was also critical to determine suitable solubilization conditions for membrane proteins during the DBCO-azide conjugation. This is likely because the hydrophobic tails of detergents/lipids may decrease the overall conjugation efficiency by interfering with the hydrophobic DBCO moiety (*Huynh et al., 2019*). Indeed, the best condition was in the absence of any detergents or lipids (*Figure 3*; no detergents/lipids only during the conjugation step). This situation is possible because membrane proteins are sparsely tethered to the chamber surface, which kept them from aggregating. However, not using detergents/lipids means that the membrane proteins are definitely deformed from their native folds. Therefore, we sought an optimal solubilization condition for membrane proteins during the DBCO-azide conjugation step. Among various concentrations of diverse detergent/lipid molecules (*Figure 3* and *Figure 3—source data 1*), we found that 0.05% (w/v) DDM detergent condition enabled the stable solubilization of membrane proteins along with the DBCO-mediated bead tethering (critical micelle concentration of DDM = ~0.01%).

The length of DNA handles also played a crucial role in the bead tethering. We failed in the tethering with short handles, typically less than 500 bp as used in single-molecule tweezers (*Choi et al., 2022*; *Moessmer et al., 2022*; *Kostrz et al., 2019*; *Min et al., 2018*; *Hao et al., 2017*; *Figure 3* and *Figure 3—source data 1*). This result may be due to the decreased radius-of-gyration of the short handles confined to the chamber surface, leading to reduced efficiency of the DBCO-azide conjugation in detergent/lipid solutions. Following a 1 hr incubation of the beads in the single-molecule chamber at 25°C, unconjugated beads were washed, and the detergent micelles were exchanged with bicelles to reconstitute the lipid bilayer environment for membrane proteins (*Figure 2D*; 1 min at 20–22°C).

## Mechanical pulling on biomolecules with opposing hydrophobicity

Our tweezer method with the strong molecular tethering allowed for repetitive mechanical pulling on the two test biomolecules (*Figure 4A and B*). We found that scTMHC2 was unfolded at 18.2±0.1 pN at a magnet speed of 0.5 mm/s (*Figure 4—figure supplement 1*), which is consistent with a previous study (*Lu et al., 2018*). Its unfolding forces and step sizes appeared in the helix-coil transition zone bounded by two hypothetical unfolded states (*Choi et al., 2019*; *Min et al., 2015 Figure 4C*) – the unfolded polypeptide state with all helices fully unstructured (the unfolded coil state; $U_c$) and the unfolded helical state with all helices fully structured ($U_h$). We modeled the two unfolded states using the worm-like chain (WLC) and Kessler-Rabin (KR) polymer models for polypeptides and helices, respectively (see Methods). The area of data points scattered around 18 pN matched that of the helix-coil transition of a helical membrane protein, GlpG (*Choi et al., 2019*; *Min et al., 2015*).

At high forces, the TM helical portions of the $U_h$ state are likely to be located at the water-membrane interface as they are highly stretched. However, at lower forces, they can be inserted into the membrane with angles (*Choi et al., 2019*). Once the TM helices become unstructured, as in the $U_c$ state, the corresponding polypeptide portions are unlikely to be within the membrane due to an energetic cost of the polar backbone in contact with nonpolar lipid tails. Thus, unfolding above the helix-coil transition zone will be a coupled event of unfolding and extraction from the membrane. It

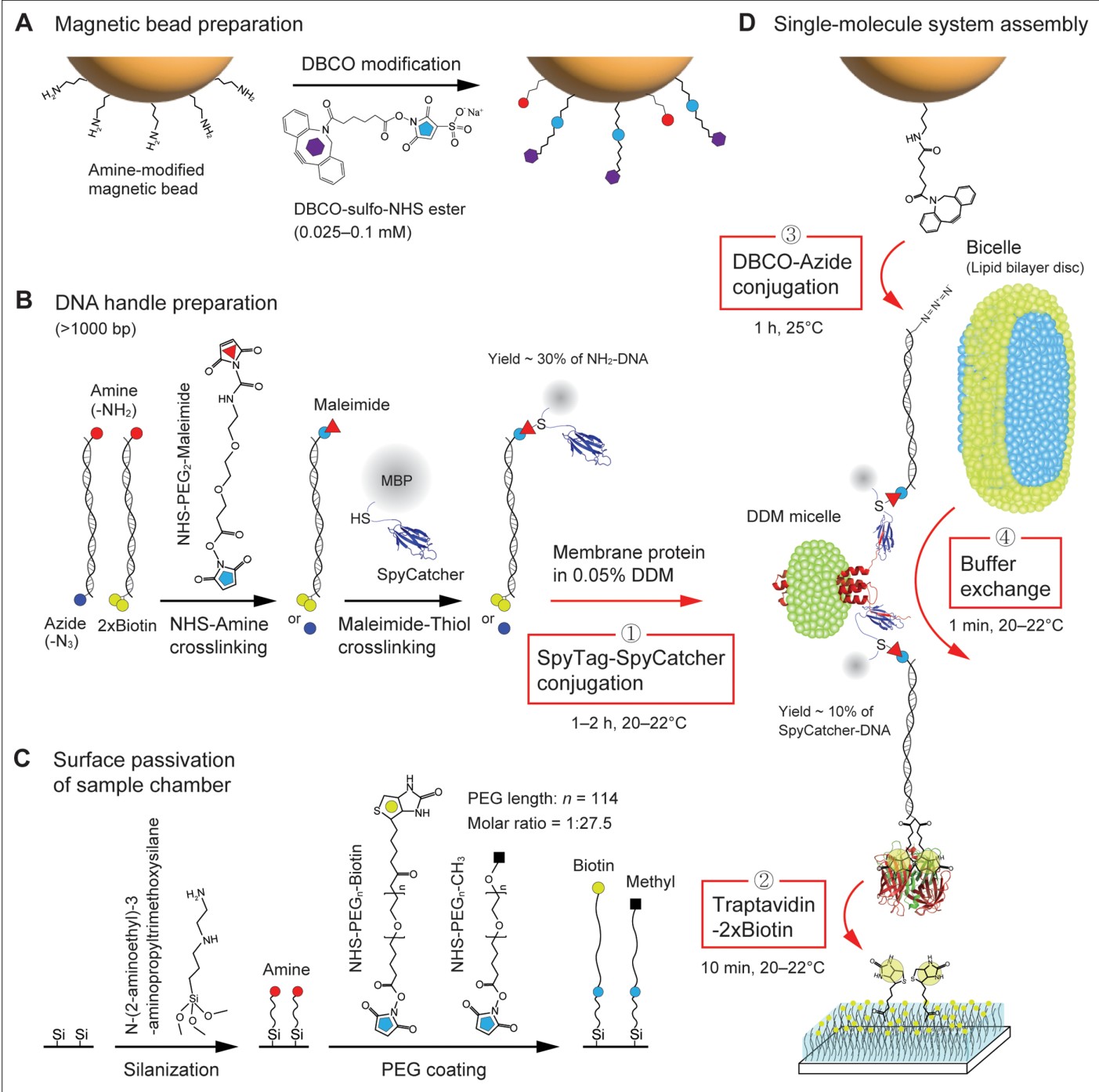

**Figure 2.** Procedure of single membrane protein tethering. (**A**) Dibenzocyclooctyne (DBCO) modification on the surface of amine-coated magnetic beads. (**B**) Preparation of SpyCatcher-DNA handles. One end of the handles is labeled with amine, while the other end of the handles is labeled with azide or dual biotins (2×biotin). The amine end is modified by maleimide, followed by the attachment to the cystine of SpyCatcher which is fused to maltose binding protein (MBP). (**C**) Passivation of sample chamber surface. The surface is functionalized by amines via silanization, followed by polyethylene glycol (PEG) coating. (**D**) Assembly of single-molecule pulling system. The SpyCatcher-DNA handles are attached to a membrane protein solubilized in 0.05% n-Dodecyl-β-Maltoside (DDM) via SpyTag/SpyCatcher binding (first step). The hybrid molecular construct is bound to traptavidin at its 2×biotin-modified end. The remaining biotin-binding pockets of the traptativin are then attached to biotins on the surface (second step). The DBCO-modified bead is tethered to the other end of the molecular construct via DBCO-azide conjugation (third step). The final buffer exchange with bicelles allows for lipid bilayer environment for the membrane protein (fourth step).

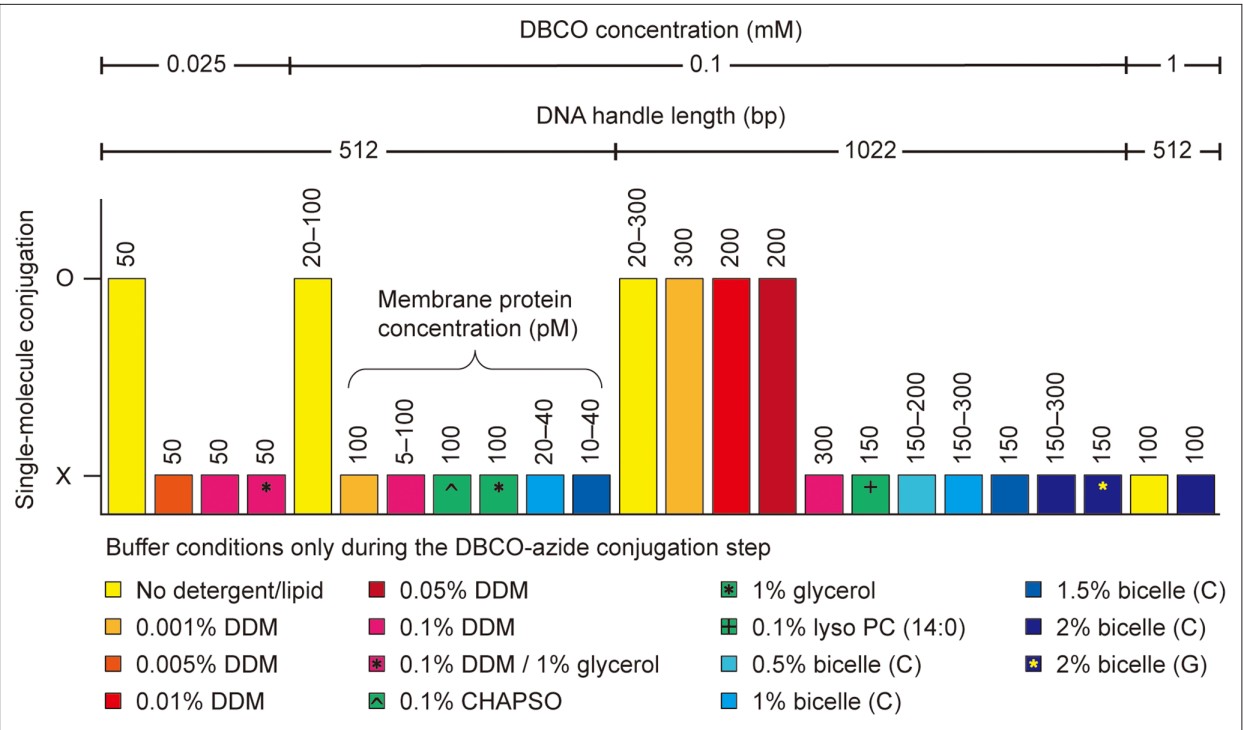

**Figure 3.** Single-molecule dibenzocyclooctyne (DBCO)-azide conjugation in detergent/lipid environments. The DBCO concentration indicates the final concentration of DBCO-sulfo-*N*-hydroxysuccinimide (NHS) ester crosslinker in DBCO modification on bead surface. The other conditions, such as DNA handle length, membrane protein concentration, and detergent/lipid concentration, indicate those in DBCO-azide conjugation step of the single-molecule system assembly. Used buffers are a phosphate-buffered saline (100 mM sodium phosphate, 150 mM NaCl, and pH 7.3) and a Tris-buffered saline (50 mM Tris, 150 mM NaCl, and pH 7.5). The bicelle (**C** or **G**) consists of DMPC (or DMPG) lipid and CHAPSO detergent at a 2.5:1 molar ratio. The % indicates w/v %. The symbol O indicates successful conjugation between DBCO-modified beads and target membrane proteins. In this case, specifically tethered beads are found, and the force-extension curves of the molecular constructs are obtained (confirmed by three replicates). The symbol X indicates unsuccessful conjugation or entire nonspecific binding of beads to the surface. In this case, no data are obtained. See *Figure 3—source data 1* for the full list. Abbreviations: DMPC, Dimyristoylphosphatidylcholine; DMPG, Dimyristoylphosphatidylglycerol; CHAPSO, 3-[(3-cholamidopropyl)dimethylammonio]-2-hydroxy-1-propanesulfonate.

The online version of this article includes the following source data for figure 3:

**Source data 1.** Full list of tested conditions for dibenzocyclooctyne (DBCO)-azide conjugation.

is still unclear, however, whether the membrane protein in the $U_c$ state is entirely in water or at the water-membrane interface. The fully unstructured protein may frequently bind and unbind from the bicelles in solution through the interactions with hydrophobic side chains.

## Assessing robustness of our single-molecule tweezers

To assess how well the tweezers could withstand the force, we first sought a practical upper bound for the force that could be applied to most biological processes. We surveyed 53 distinct proteins studied by optical/magnetic tweezers. For 98% of them, the most probable or average unfolding forces were measured to be less than 50 pN, regardless of their molecular weights and used force-loading rates (*Figure 4D* and *Figure 4—source data 1*). Since protein folding involves numerous noncovalent inter-actions, ranging from hundreds to thousands of inter-residue interactions (*Gromiha and Selvaraj, 2004*), most biomolecular interactions could be studied below the 50 pN. Thus, the 50 pN was set as an upper bound for our robustness test. We measured the survival probability of our linkage system over time at the upper-bound force and estimated the mean lifetime until bead detachment to be 11.7±1.5 hr (*Figure 4E*). In contrast, the median lifetime of a conventional linkage system that uses 1 dig and 1 biotin is very short, around 7 min at 45 pN (*Janissen et al., 2014*). This result demonstrates that our system is at least 100 times more stable than the conventional system used in the previous membrane protein studies.

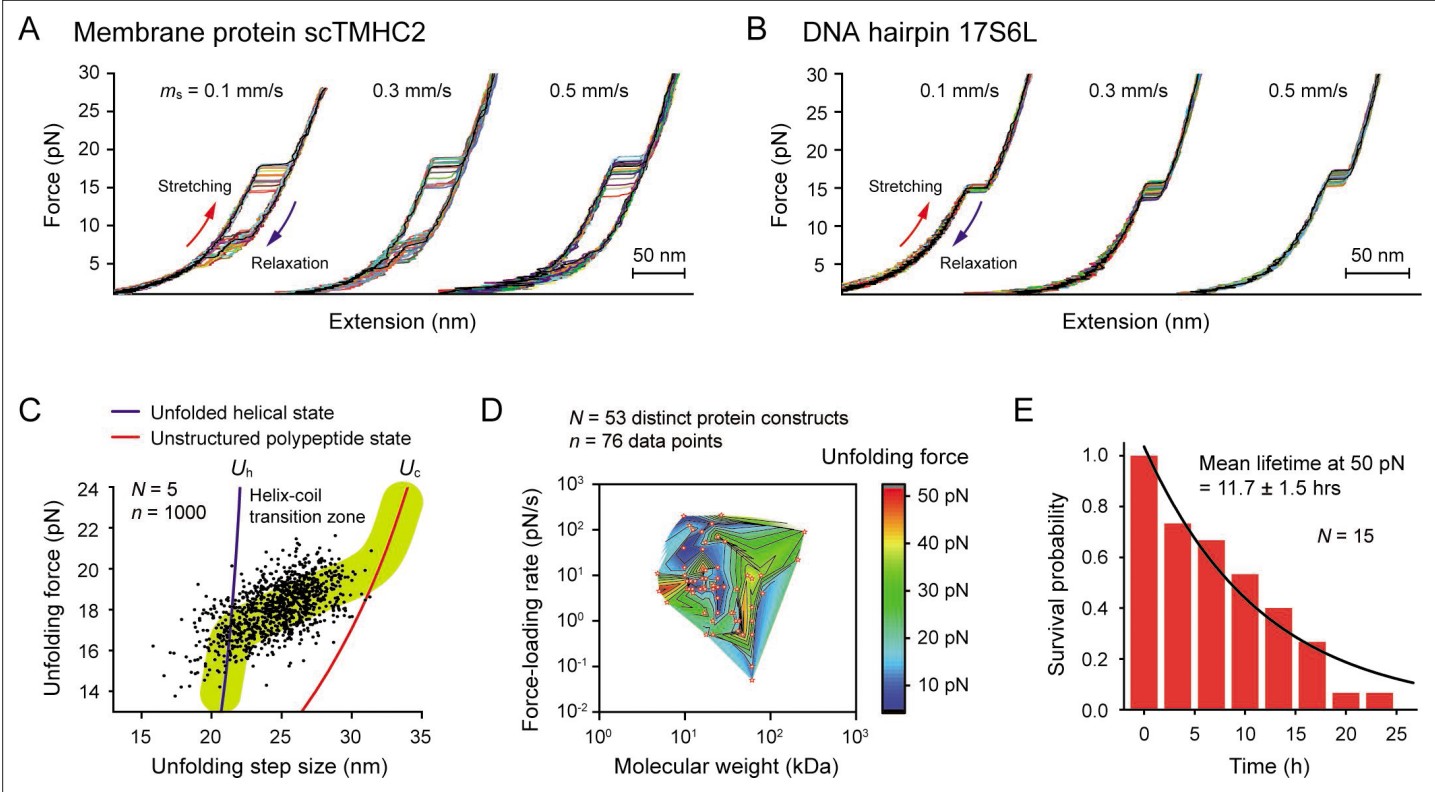

**Figure 4.** Robustness evaluation of our single-molecule tweezers. (**A and B**) Representative force-extension curves of the membrane protein scTMHC2 (**A**) and the DNA hairpin 17S6L (**B**) at different magnet speeds (N=20 for each condition). The magnet speed is denoted as $m_s$. (**C**) Scatter plot of unfolding forces and step sizes of the membrane protein (N=5 molecules, n=1000 data points, and $m_s$ = 0.5 mm/s). The blue and red lines represent the protein force-extension curves for the unfolded helical state ($U_h$) and the fully, unstructured polypeptide state ($U_p$), respectively. (**D**) Survey of protein unfolding forces that were previously measured by single-molecule tweezers (N=53 distinct protein constructs, n=76 data points). See *Figure 4—source data 1* for the full list. (**E**) Survival probability of our tethering system as a function of time. The mean lifetime of 11.7±1.5 (SE) hr at 50 pN was obtained by fitting the plot with exponential decay function (N=15).

The online version of this article includes the following source data and figure supplement(s) for figure 4:

**Source data 1.** Survey of protein unfolding forces in optical and magnetic tweezers.

**Figure supplement 1.** Force calibration and unfolding forces/sizes of scTMHC2.

## Robust single-molecule tweezers capable of 1000 pulling cycles

Our tweezer method can also perform a large number of unfolding cycles with a single membrane protein, up to a thousand times on average (*Figure 5A*). The measured variables for individual molecules, such as the unfolding force and step size, appear consistent over the thousand cycles within inherent stochastic nature of protein (un)folding (*Figure 5—figure supplement 1*). However, the values are affected by the force calibration error due to variations in bead size (~3% in diameter; Methods). Thus, to effectively exclude the bead size effect, we averaged the values for different molecules tethered to different beads at each pulling cycle (*Figure 5B*). We then tracked the level of mean forces/sizes to assess the measurement stability over the thousand cycles. The relative SD (RSD; defined as the ratio of the SD to the mean) was estimated to be 7.0 and 10.4% (*Figure 5B*, right), mainly reflecting the fluctuations from the stochastic barrier crossing. Nonetheless, the level of the mean values remained almost constant over the thousand times pulling (slope ~$10^{-4}$ pN[nm]/cycle; *Figure 5—figure supplement 2* for the DNA hairpin). This result demonstrates that our method is also stable over the large number of stretching experiments. A systematic trend between the unfolding force/size and the progress of pulling cycles may indicate a temporal alteration of the protein fold, such as protein aging effect via oxidative damage. Indeed, single-molecule tweezers have revealed the phenomenon of the protein aging or folding fatigue of several soluble proteins (*Valle-Orero et al., 2017*; *Kellermayer et al., 2001*). Thus, our method offers the ability to quantify the oxidative

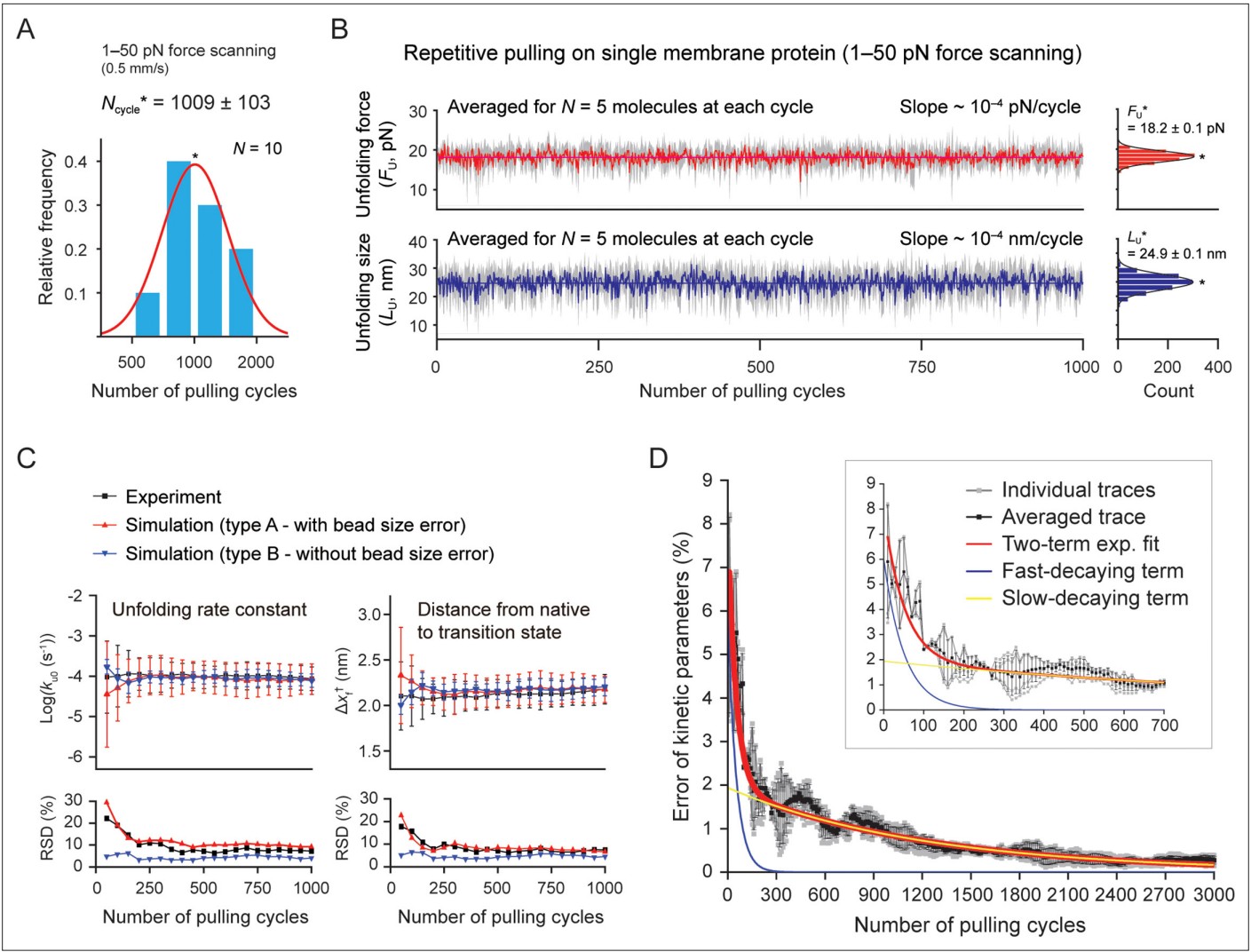

**Figure 5.** Stable single-molecule tweezers over a thousand pulling cycles. (**A**) Applied number of pulling on single membrane protein until bead detachment (N=10 molecules). The peak value ($N_{cycle}$*) is 1009±103 (SE). The force-scanning range was 1–50 pN. The magnet speed was 0.5 mm/s. (**B**) Unfolding force and step size during progress of pulling cycles (analyzed for five molecules surviving more than the thousand cycles). The colored traces represent the averaged traces for five different molecules at each cycle. The SD at each cycle is denoted in gray. The slope of the traces was obtained as ~$10^{-4}$ pN(nm)/cycle by linear fitting. Count histograms of the values are shown on the right (peak value ± SE). (**C**) Kinetic parameters during progress of pulling cycles (N=5 molecules). $k_{u0}$ and $\Delta x_f^{\dagger}$ indicate the unfolding rate constant at zero force and the distance between the native and transition states, respectively. Relative SD (RSD) indicates the ratio of SD to mean. (**D**) Error of kinetic parameters during progress of pulling cycles. The gray traces indicate individual traces created from Monte Carlo (MC) simulations shown in ***Figure 5—figure supplement 5***. The black trace is the trace averaged for the individual traces. The red curve represents the two-term exponential fitting. The blue and yellow curves represent the fast- and slow-decaying terms, respectively. The inset shows the initial hundreds of pulling cycles.

The online version of this article includes the following source data and figure supplement(s) for figure 5:

**Source data 1.** Codes for the Monte Carlo simulations.

**Figure supplement 1.** Unfolding force/size of individual proteins during progress of pulling cycles.

**Figure supplement 2.** Unfolding/refolding forces during progress of pulling cycles for DNA hairpin.

**Figure supplement 3.** Unfolding/refolding probabilities as a function of force.

**Figure supplement 4.** Comparison of two modes of kinetic analyses.

**Figure supplement 5.** Error estimation of kinetic parameters.

**Figure supplement 6.** Comparison of two regression analysis models.

damage on membrane protein folds and investigate how chaperone proteins or ligands interact with the changing folds over time.

## Assessing statistical reliability of pulling-cycle experiments

The pulling-cycle approach can determine important physicochemical variables, such as the kinetics and free energy changes for molecular transitions (*Wang et al., 2022*; *Janovjak et al., 2008*). To estimate their errors during the progress of pulling cycles, we performed Monte Carlo (MC) simulations and compared the results with experimental data (*Figure 5C and D* and *Figure 5—figure supplements 3–6*; Methods). We conducted two types of simulations to consider the bead size effect – type A and B with or without the bead size error. For this estimation, we examined the kinetic parameters ($k_{u0}, \Delta x_f^\dagger$) for the coupled event of unfolding and extraction, which could be obtained from the unfolding probability as a function of force (*Min et al., 2015*; *Figure 5—figure supplement 3* and Methods). We averaged the values for five molecules at every 50 pulling cycles (*Figure 5C*). As expected, the RSD for the experiment and type A simulation were a few to tens of % higher than those of the type B simulation (*Figure 5C*, lower). However, the mean values of the kinetic parameters converged with one another after the initial hundred cycles, indicating that the bead size effect is effectively eliminated by the averaging (*Figure 5C*, upper). The kinetic error trend was decomposed into the fast and slow decays, with $\tau$=52 and 1223 cycles, respectively (*Figure 5D* and Methods). With correction of instrumental errors (*Jacobson and Perkins, 2020*; *Cossio et al., 2015*; *Neupane and Woodside, 2016*), the error of kinetics can be approximated to be 3.7, 1.8, and 0.9% at 60, 200, and 1000 pulling cycles, respectively. Therefore, our method is statistically reliable in obtaining the kinetic parameters with sufficiently small errors in a practical sense (see Methods for more details).

## Structural transitions of a membrane protein over extended time scales

Using the robust membrane protein tweezers, we attempted to observe long-lasting, repetitive (un)folding transitions at a constant force, which may be more difficult for membrane proteins. All membrane proteins studied by single-molecule tweezers exhibit relatively large mechanical hysteresis with tens-of-pN gaps between their unfolding and refolding forces, even at very low loading rates like 1 pN/s (*Choi et al., 2019*; *Choi et al., 2022*; *Min et al., 2018*; *Min et al., 2015*; *Lu et al., 2018*). This large mechanical hysteresis hinders the observation of rapid transitions at a constant force. Recent studies have managed to capture the transitions for short time scales of tens-to-hundreds of seconds at 5–7 pN (*Choi et al., 2019*; *Choi et al., 2022*). However, the force level is much more favorable to the native states. As a result, once the proteins have folded, it is unlikely to unfold again within the practical time scales of experiments.

In light of this, upon complete unfolding of scTMHC2 to the $U_c$ state above the helix-coil transition zone, we decreased the force to 12 pN, which is close to the mean force of the most probable unfolding and refolding forces (~13 pN; *Lu et al., 2018*). The high stability of our tweezers allowed us to observe persistent (un)folding transitions lasting for a long time of 9 hr (*Figure 6A*). The extension distribution displayed four clear peaks (*Figure 6A*, right), indicating that there are four major structural states. We analyzed the interconversions between the states using the HMM analysis (*Zhang et al., 2016*; *Jiao et al., 2017*).

The lowest extension state corresponds to the fully folded state (*N*), and the upper extension states correspond to the partially or fully unfolded states. Using the WLC and KR polymer modeling at 12 pN (*Figure 6B*), we identified the corresponding unfolded states that match the measured extension levels (*Figure 6C*). The uppermost extension state was located 27.9±1.3 nm away from the lowest extension state. This extension level more matches the estimated value of 25.3 nm for the $U_c$ state than 20.5 nm for the $U_h$ state. It should be pointed out that scTMHC2 was designed as a single-chained homodimer, which consists of two identical monomers with two helices connected by a short peptide linker (*Lu et al., 2018*; *Figure 1—figure supplement 1A* for its amino acid sequence). Indeed, the extension levels for the middle two states, 11.3±1.7 and 22.6±2.1 nm, correspond well with the estimated values of 10.3 and 20.5 nm for the intermediate helical states with one monomer unfolded and the other folded ($I_{1,2}$) and the $U_h$ state, respectively (*Figure 6B and C*). The $I_1$ and $I_2$ states are indistinguishable as their step sizes are identical.

To reconstruct the folding energy landscape ($\Delta G[l]$) at 12 pN, we employed a nonlinear constrained iterative deconvolution method (*Gebhardt et al., 2010*; *Woodside et al., 2006*), which is independent

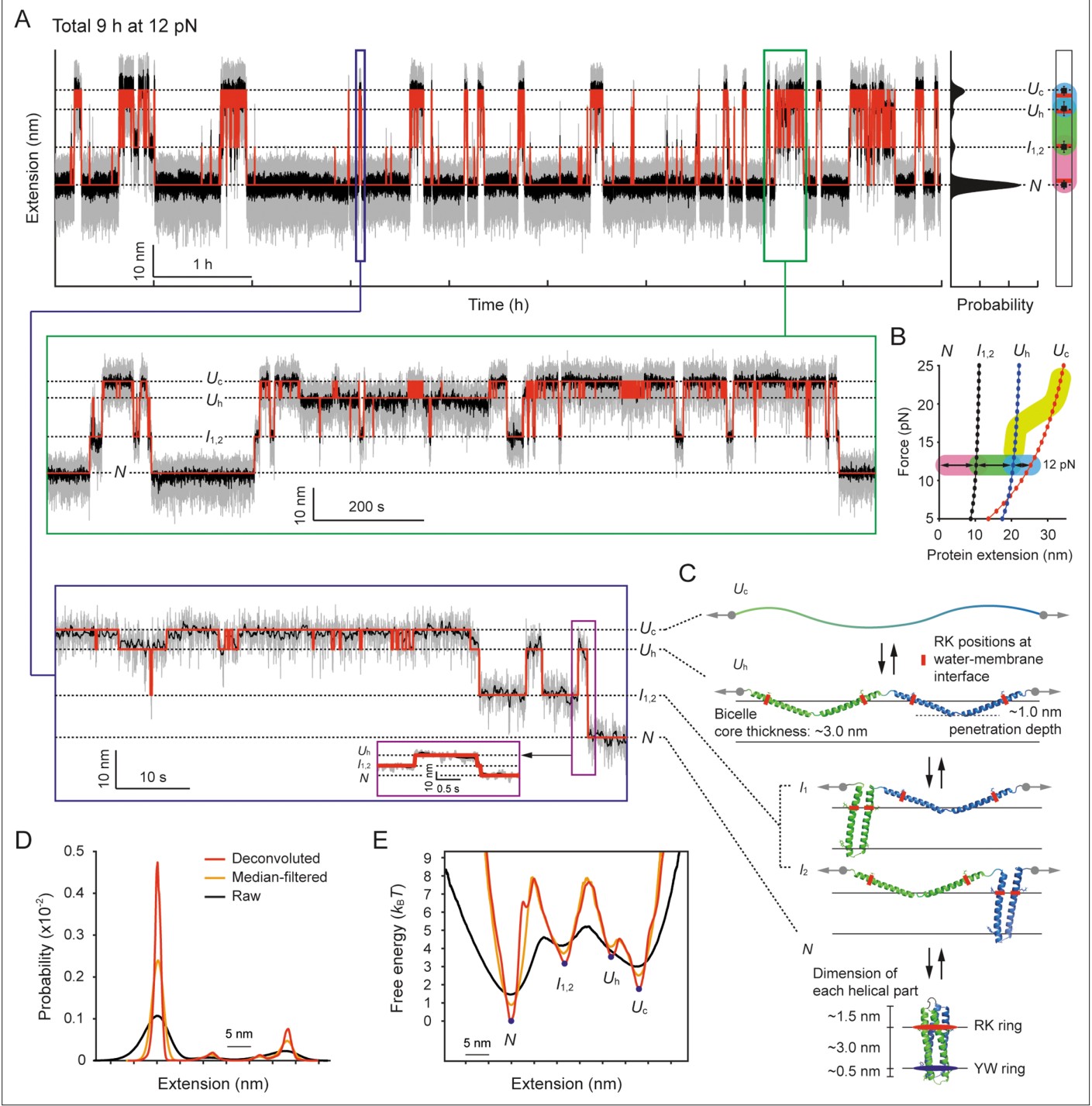

**Figure 6.** Structural transitions of scTMHC2 for 9 hr at 12 pN. (**A**) Time-resolved extension trace of scTMHC2. The gray and black traces are the raw trace and median-filtered trace with 4.5 Hz window, respectively. The red lines indicate the state positions determined by the hidden Markov modeling (HMM) analysis. The probability density of the extension is shown on the right. The peak positions (black dots) with error bars (SD) are compared with the polymer model-estimated positions for each state (red lines). The color code for each transition is same as in panel B. The below insets are the zoomed-in plots for corresponding time regions. (**B**) Protein force-extension curves estimated by the worm-like chain (WLC) and Kessler-Rabin (KR) models. The curves for each state are denoted as the colored lines with dots. The transitions between the states at 12 pN are shown as arrows with colored backgrounds (pink, green, and blue). The yellow thick curve represents the helix-coil transition zone in the stretching experiments shown in *Figure 4C*. (**C**) Predicted structural states of scTMHC2 for the four extension positions at 12 pN. The RK and YW ring were designed to be positioned at the water-membrane interfaces. The gray arrows indicate the direction of applied force to the protein. The black arrows represent the allowed

*Figure 6 continued on next page*

*Figure 6 continued*

transitions between the states. (**D**) Probability density of protein extension obtained using the deconvolution method. (**E**) Folding energy landscape obtained from the deconvoluted probability density and Boltzmann relation.

The online version of this article includes the following figure supplement(s) for figure 6:

**Figure supplement 1.** Evaluation of our deconvolution method using simulated traces.

**Figure supplement 2.** Folding and unfolding times of scTMHC2 at 12 pN.

**Figure supplement 3.** Average penetration angle and depth of helices of the $U_h$ state at 12 pN.

of the kinetic models (*Figure 6—figure supplement 1* for its test with simulated data; see Methods). This approach eliminates the fluctuation noise of the bead and handles by iteratively deconvoluting the initial probability density of the measured extension ($P[l]$). Using this approach, we obtained the probability density of the protein extension ($p[l]$; *Figure 6D*). We then characterized the $\Delta G(l)$ using the Boltzmann relation, $\Delta G(l) = -k_B T \cdot \ln(p[l]/p[l=0])$ (*Figure 6E*). We also separately extracted the observed (un)folding times ($\tau_{obs}$) between the states using the HMM and dwell time analysis (*Zhang et al., 2016*; *Jiao et al., 2017*; *Figure 6—figure supplement 2*). The observed times were corrected for the limited temporal resolution and tethered bead-handle effect (*Jacobson and Perkins, 2020*; *Cossio et al., 2015*; $\tau_{(un)folding}$; see Methods).

## Mechanistic dissection of the folding transitions

At the constant 12 pN, we observed rapid interconversions between the $U_h$ and $U_c$ states, i.e., the helix-coil transitions (see the zoomed-in traces in *Figure 6A*). The $U_c$ state is slightly more favorable than the $U_h$ state ($\Delta G_{U_c\text{-}U_h} = -1.7\ k_B T$), although this force is ~6 pN lower than the ~18 pN force level of the helix-to-coil transition during the stretching experiments (*Figure 4C*). This hysteresis effect arises likely because the first unfolding transition ($N$-to-$I_{1,2}$) is the rate-limiting step. Even though the force application of 12 pN lowers the energy barrier for this transition, the barrier (rate) is still highest (slowest) among all transitions ($\Delta G^{\dagger}_{N\text{-to-}I} = 7.8\ k_B T$ and $\tau_{N\text{-to-}I} = 381$ s). Moreover, only the transitions between adjacent states are allowed such as $N \leftrightarrow I_{1,2} \leftrightarrow U_h \leftrightarrow U_c$ (*Figure 6—figure supplement 2*). Thus, in the stretching experiment where force is increased over time (1.4 pN/s at 12 pN), the force quickly exceeds 12 pN before the rate-limiting transition is driven. The final helix-to-coil transition follows at higher forces, resulting in the mechanical hysteresis.

While the random coil of the $U_c$ state would be extracted out of the membrane as discussed in a previous section, the helical domains of the $U_h$ state would penetrate into the membrane by an average of ~1.0 nm at 12 pN (*Figure 6C* and *Figure 6—figure supplement 3*; Methods). This penetration covers approximately one third of the bicelle membrane core of ~3.0 nm thickness (*Leite et al., 2022*; *Giudice et al., 2022*; *Murugova et al., 2022*; Methods), similar to the so-called zigzag state defined in a previous study (*Choi et al., 2019*), where a similar penetration depth of ~1.0 nm was estimated at 8 pN for GlpG. Given that the penetration depth varies depending on structural factors, such as the lengths of helices and linkers, our estimation is considered to be reasonable (see Methods). This estimation suggests that the helix-coil transitions likely accompany the insertion and extraction of TM helices in and out of the bicelle membrane.

The four helices connected by peptide linkers can undergo the transition independently, although we observed the cooperative transitions of all four helices. The transition of one part may energetically benefit adjacent parts for the same state by the geometrical perturbation of the accompanying insertion/extraction event (as shown in *Figure 6C*). It is also possible that, due to a limited spatiotemporal resolution, we could not detect possible intermediate states during the transitions between the $U_c$ and $U_h$ states.

The folding pathway observed at 12 pN follows a strict progression from the $U_c$ to $U_h$ to $I_{1,2}$ to $N$ state, with no bypass of the middle $U_h$ and $I_{1,2}$ states. Thus, the $U_h$ state evidently serves as the primed state for the first helix-helix association of the $U_h$-to-$I_{1,2}$ transition (*Figure 6C*). The final $I_{1,2}$-to-$N$ transition involves two events, i.e., the second helix-helix association and the dimerization of two helical hairpins. At 12 pN, two unassembled helices are separated by ~11 nm, whereas two helical hairpins that have not yet dimerized are very close with only ~1 nm by the intervening peptide linker. As a result, owing to the energetic penalty by the ~10 times greater distance, the helix-helix association

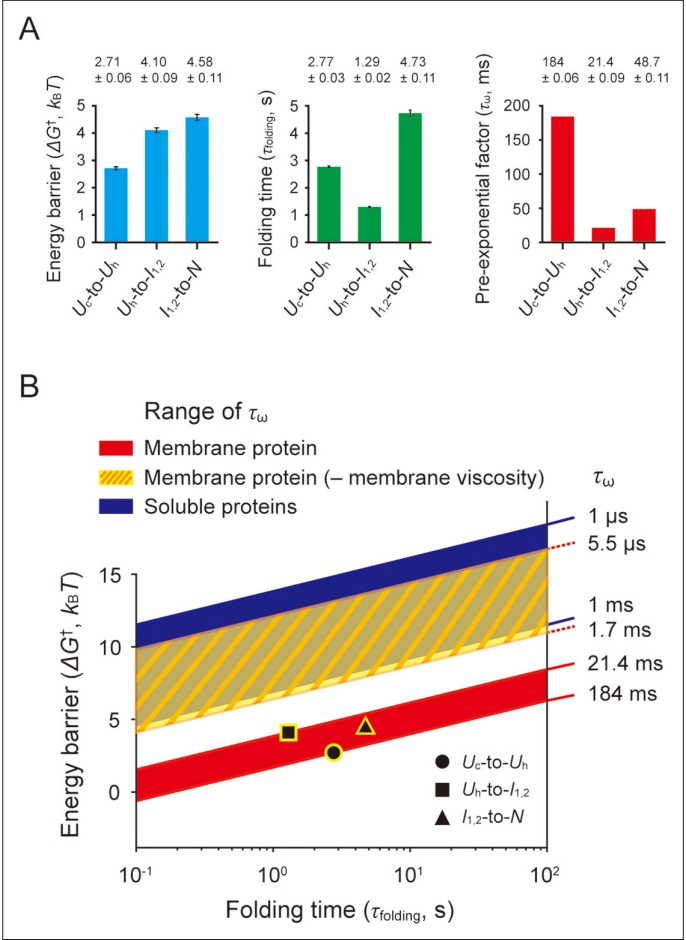

**Figure 7.** Estimation of pre-exponential factors for scTMHC2 folding transitions. (**A**) Energy barriers, folding times, and pre-exponential factors of the transitions. The energy barrier heights ($\Delta G^\dagger$) were obtained by the deconvolution method. The folding times ($\tau_{folding}$) were obtained by the hidden Markov modeling (HMM) analysis and instrumental error correction. The two measured parameters yield the pre-exponential factors ($\tau_\omega$) of the Kramers equation for the folding transitions. The error bars represent SE. (**B**) $\Delta G^\dagger$–$\tau_{folding}$ plane showing the regions for designated $\tau_\omega$ ranges. The black data points indicate the measured values of $\Delta G^\dagger$ and $\tau_{folding}$. The red-shaded area, yellow-shaded area with slash lines, and blue-shaded area correspond to the regions for the estimated $\tau_\omega$ range, the $\tau_\omega$ range in the absence of lipid membrane viscosity (denoted as '– membrane viscosity'), and the $\tau_\omega$ range observed for soluble proteins, respectively.

The online version of this article includes the following source data for figure 7:

**Source data 1.** Survey of measured viscosity for DMPC lipid membranes.

is likely the rate-limiting step than the dimerization. Indeed, the energy barriers for the $U_h$-to-$I_{1,2}$ and $I_{1,2}$-to-$N$ transitions differ by only 0.5 $k_B T$, which is less than the thermal energy level (**Figure 6E**).

At 12 pN, a helical hairpin is marginally stable ($\Delta G_{I-U_h} = -0.3\ k_B T$), whereas the finally folded state (the dimerized state of two helical hairpins) is ~10 times more stable ($\Delta G_{N-I} = -3.2\ k_B T$). This indicates that the formation of the other helical hairpin alone does not significantly contribute to the native fold stability. Instead, the final dimerization is likely the primary stabilizing step. This also reconciles the steep free energy slope down the small bump observed between the $I_{1,2}$ and $N$ states (**Figure 6E**), which may involve the final dimerization step. Once finally folded, the ~3.0 nm TM helical part bounded by the RK and YW rings well matches the ~3.0 nm thickness of the bicelle membrane core (**Leite et al., 2022**; **Giudice et al., 2022**; **Murugova et al., 2022**; **Figure 6C** and Methods).

## Estimating the speed limits of the folding transitions

From the characterized folding energy barriers ($\Delta G^\dagger$) and folding times ($\tau_{folding}$), we obtained the pre-exponential factor ($\tau_\omega$), an estimate of the folding 'speed limit' (**Kubelka et al., 2004**; **Chung and**

*Eaton, 2018*), in the Kramers rate theory ($\tau_{folding} = \tau_\omega \cdot \exp[\Delta G^\dagger / k_B T]$; *Hänggi et al., 1990*; *Figure 7A* and Methods). Here, $\tau_\omega$ is $2\pi\gamma/\omega_i\omega_t$ in a high frictional regime such as protein folding (*Hagen, 2010*), $\omega_i$ and $\omega_t$ are the angular frequencies that characterize the curvature of energy well and barrier top, and $\gamma$ is the friction coefficient. This approach is a modification of a previous study where the frequency factor ($k_\omega = 1/\tau_\omega$) was estimated for a soluble protein (*Gebhardt et al., 2010*).

To this end, it is necessary to acquire a large amount of extension data during the transitions using a high temporal resolution. However, the temporal resolution of typical magnetic tweezers is inherently limited due to the camera-based method for the bead position tracking, which restricts the speed limit estimation. Our robust single-molecule method overcomes this limitation by conducting the prolonged observation of numerous transitions, resulting in the collection of abundant data points for each transition event. This experimental strategy would not be feasible with the previous magnetic tweezer method for membrane proteins, due to the less stable molecular tethering under applied forces.

The estimated $\tau_\omega$ values for each structural transition are as follows: 184±0.06 ms for the coil-to-helix transition, 21.4±0.09 ms for the helical hairpin formation, and 48.7±0.11 ms for the final folding (helical hairpin formation plus dimerization). The range of 21–184 ms for $\tau_\omega$ is considerably lower than the range of 1 μs – 1 ms for typical soluble proteins (>100 residues; *Gebhardt et al., 2010*; *Kubelka et al., 2004*; *Hagen et al., 1996*; *Schuler et al., 2002*; *Kubelka et al., 2006*; *Dumont et al., 2009*; *Chung and Eaton, 2013*; *Szczepaniak et al., 2019*; see *Kubelka et al., 2004* for other relevant literatures). The exceedingly low speed limits for the transitions could be due to the effect of very high viscosity of lipid membranes. In the limit of a high frictional regime, $\tau_\omega$ varies linearly with the friction coefficient $\gamma$, which reflects the external friction like solvent viscosity ($\eta_s$) and the internal friction ($\eta_i$) like dissipative intrachain interactions, concerted dihedral angle rotations, and non-native salt bridges (*Hagen, 2010*; *Soranno et al., 2017*; *Chung et al., 2015*; *Qiu and Hagen, 2004*). In extremely high viscous environments, the solvent viscosity become dominant for the friction term (*Hagen, 2010*; *Eaton, 2021*; *Ansari et al., 1992*), leading to the relationship of $\tau_\omega \propto \gamma \propto \eta_s$. This approximate linear relationship between $\tau_\omega$ and $\eta_s$ serves as a working hypothesis that can explain why our estimated $\tau_\omega$ time scale is in stark contrast to that of soluble proteins.

Indeed, the viscosity of DMPC lipid membranes used in this study has been measured using various methods to be approximately $70$–$2.3\times10^4$ times higher than that of water (*Nagao et al., 2017*; *Nojima and Iwata, 2014*; *Wu et al., 2013*; *Dimova et al., 2000*; *Bahri et al., 2005*; *Filippov et al., 2003*; *Figure 7—source data 1* for the measured viscosity values). If we exclude the smallest and largest values to avoid potential artifacts, the range of the viscosity ratio between the lipid membranes and water is approximately $1.1\times10^2$–$3.9\times10^3$. With this range of the viscosity ratios, we determined a maximum possible $\tau_\omega$ range of 5.5 μs – 1.7 ms in the absence of the membrane viscosity. This rough estimate largely overlaps with the $\tau_\omega$ range for soluble proteins (*Figure 7B*), providing a tantalizing clue for the origin of the slow speed limits, i.e., the viscosity of lipid membranes. We note that the viscosity of lipid membranes was not measured in bicelles but rather in various other structures such as small to giant unilamellar vesicles and supported membranes, which could introduce possible errors into our analysis.

A helical hairpin is the minimal structural size for helical membrane proteins that require the tertiary folding (*Choi et al., 2019*; *Corin and Bowie, 2022*; *Krainer et al., 2018*; *Engelman and Steitz, 1981*), although larger membrane proteins can fold via accrual of smaller structural units on the initial template (*Choi et al., 2022*; *Corin and Bowie, 2022*). In addition, the solvent friction is expected to be the primary determinant for $\tau_\omega$ in highly viscous lipid membrane environments (*Ansari et al., 1992*). Thus, it is tempting to speculate that a time scale of ~20 ms for the helical hairpin formation may be around the upper limit of folding speeds for the entire helical membrane proteome.

The coil-to-helix transition has a time scale of ~180 ms for $\tau_\omega$, which is ~9 times slower than the helical hairpin formation. The slowing effect may be due to the transition itself that is a totally different process and/or the accompanying helix insertion into membranes. The final folding step, i.e., the helical hairpin formation followed by dimerization, has a slightly slower time scale of ~50 ms for $\tau_\omega$. This slowing effect may be due to the additional dimerization step and/or the slight extraction of the unassembled helices of the $I_{1,2}$ state from the membrane (as shown in *Figure 6C*).

## Discussion

We established the most robust tweezer method for membrane protein studies, but there are already a variety of stable tweezer methods for their own purposes. A covalent DNA tethering with thiol-maleimide and amine-carboxyl linkages exhibited a minimum lifetime of ~23 hr at 45 pN (*Janissen et al., 2014*). Another DNA tethering method using multiple biotins and DBCOs also showed a high stability with a mean lifetime of ~60 hr at 45 pN (*Lin et al., 2023*). The linkages of thiol-maleimide and biotin-streptavidin variants were utilized for a multiplexed protein unfolding study (*Löf et al., 2019*). Although the anchoring geometry of the biotin-streptavidin tethering affected the lifetime distributions (*Löf et al., 2019*; *Gruber et al., 2020*), the mean lifetime of the slowest kinetics was measured as ~19 hr at 45 pN (*Löf et al., 2019*). This approach observed protein (un)folding transitions for 6 days at 7.3 pN. Other covalent methods using HaloTag and streptavidin (*Popa et al., 2016*; *Chen et al., 2015*; *Tapia-Rojo et al., 2023*) can remain stable for 2 weeks. However, the time span was the total duration for an unfolding-pulse experiment where the protein was subjected to a force jump to 45 pN for 45 s once a day, while being held at 4.3 pN throughout the days (*Popa et al., 2016*). The same tethering approach observed protein (un) folding transitions for ~8 hr at 4.5 pN (*Chen et al., 2015*) and ~14 hr at 8.5 pN (*Tapia-Rojo et al., 2023*).

All three widely used force-dependent kinetic models assume that the applied force is independent of the pre-exponential factor as in $\tau(F) = \tau_\omega \cdot \exp(-\Delta G^\dagger[F]/k_BT) = \tau_\omega \cdot f(\Delta G^\dagger_0, F, \Delta x)$, where $\Delta G^\dagger_0$ is the energy barrier height at zero force, $F$ is the applied force, and $\Delta x$ is the distance to the transition state (*Hane et al., 2014*). Additionally, the deeper insertion of helices into the membrane at lower forces is effectively the rotation of already-inserted tilted helices toward the upright position. The helix rotation event also occurs at zero force, as the unassembled helices are tilted in a zigzag manner by the geometrical constraints at the intervening linkers. Thus, the force-induced rotation of helices within the membrane does not involve different kinds of transitions, which likely manifest as an elevation of energy barrier heights. However, this argument starts to break down as the soluble part on the membrane side opposite to the protein pulling side becomes dominant, which involves the transfer event between the membrane and water. In our case, the soluble portion is ~23% of the TM helical part in term of the number of amino acid residues. Therefore, the estimated $\tau_\omega$ values for each transition would be valid at zero force within the error due to the transfer event.

Recently, Choi et al. measured the rate constants for helical hairpin formations in GlpG within bicelle membranes at low forces of 5–7 pN (*Choi et al., 2019*). By extrapolating to zero force using Dudko-Hummer-Szabo model, the authors estimated the $k_0$'s corresponding to approximately 110–140 ms. These values indeed do not exceed the ~20 ms speed limit for a helical hairpin formation we estimated here. A high-speed atomic force microscopy with ~1 μs temporal resolution observed μs-rapid (un)folding transitions between numerous intermediates for bacteriorhodopsin (*Yu et al., 2017*). However, the rapid transitions involve the coupled insertion and folding of a few amino acid residues from the fully unstructured polypeptide that is entirely exposed to an aqueous solution. Therefore, the μs time scales for the conformational changes are not relevant to the speed limit of membrane protein folding within lipid bilayers.

The significantly lower speed limits of the structural transitions arise likely from the high viscosity of lipid membranes. This hypothesis reconciles the previous studies on the speed limit of soluble protein folding based on the Kramers theory. In the limit of high solvent friction, the approximate linear viscosity dependency is also expected to hold for a more general Grote-Hynes theory, which incorporates time-dependent friction (*Dupuis et al., 2018*). The high membrane viscosity may also be the microscopic origin for the prevalent mechanical hysteresis observed in membrane proteins – to drive the tardy transitions, you should apply higher forces for unfolding and lower forces for refolding. Simulation studies with soluble proteins have shown that changing solvent viscosity can alter folding pathways (*Klimov and Thirumalai, 1997*; *Rhee and Pande, 2008*), which may also be the case for membrane protein folding. However, it would be challenging to observe experimentally as we must selectively control the membrane viscosity without impacting other properties.

The range of $\tau_\omega$ values estimated for a helical membrane protein corresponds to very low level of the frequency factor, $k_\omega$, ranging from 5.4 s$^{-1}$ to 46.7 s$^{-1}$, sharply contrasting to the range of $10^3$–$10^6$ s$^{-1}$ for soluble proteins. This implies that the previous characterization of folding energy landscapes of helical membrane proteins (*Choi et al., 2019*; *Min et al., 2015*; *Lu et al., 2018*) was largely overestimated

by 3–12 $k_BT$ due to the simple use of the $k_\omega$ ranges for soluble proteins (*Figure 7B*). Our estimates can be used to obtain more accurate parameters for membrane protein folding.

## Conclusion

In this study, we established a robust magnetic tweezer approach that is ~100 times more stable than the conventional method used for membrane protein studies. Our method is highly stable for both constant-force and pulling-cycle experiments. Using this method, we explored the entire folding transitions of a designer membrane protein, from the fully unstructured state to the native state. By characterizing the folding energy barriers and folding times for the structural transitions observed during a 9-hr long time, we estimated the speed limit of helical membrane protein folding. Our results revealed that the folding speed limit is exceedingly low compared to soluble proteins.

## Methods

### Traptavidin purification

Traptavidin with 6xHis tag at the C-terminus encoded in pET24a vector was transformed into *Escherichia coli* BL21-Gold(DE3)pLysS (Agilent). A selected colony from a transformed agar plate was grown in 1 l of Luria-Bertani (LB) medium with 25 mg/ml kanamycin at 37°C. 0.5 mM isopropyl β-D-1-thiogalactopyranoside (IPTG) was added at $OD_{600}$ = ~0.9 to overexpress the protein for 4 hr at 37 °C. The cell culture was centrifuged at 5700 rpm for 10 min at 4°C, and the cell pellet was resuspended in 50 ml lysis buffer (10 mM $Na_2HPO_4$, 2 mM $KH_2PO_4$, pH 7.4, 287 mM NaCl, 2.7 mM KCl, 5 mM EDTA, and 1% Triton X-100). The resuspension was lysed with Emulsiflex C3 high pressure homogenizer (~17,000 psi, Avestin). The cell lysate was centrifuged at 18,000 rpm for 30 min at 4°C, the supernatant was removed, and the inclusion body pellet was washed with 10 ml extraction buffer (10 mM $Na_2HPO_4$, 2 mM $KH_2PO_4$, pH 7.4, 137 mM NaCl, 2.7 mM KCl, and 0.5% Triton X-100). The washing step was repeated three times. The washed pellet was dissolved in 6 M guanidine hydrochloride, followed by dilution into a refolding buffer (10 mM $Na_2HPO_4$, 2 mM $KH_2PO_4$, pH 7.4, 137 mM NaCl, 2.7 mM KCl, and 10 mM imidazole) and then incubation for 16 hr at 4°C (*Chivers et al., 2010*; *Howarth and Ting, 2008*). The sample was centrifuged at 17,700 *g* for 15 min at 4°C. The supernatant was incubated with 2 ml Ni-IDA resin (Takara Bio) for 2 hr at 4°C and then washed in a gravity column with 10 ml wash buffer (10 mM $Na_2HPO_4$, 2 mM $KH_2PO_4$, pH 7.4, 287 mM NaCl, 2.7 mM KCl, and 30 mM imidazole) three times. The protein sample was eluted with an elution buffer (10 mM $Na_2HPO_4$, 2 mM $KH_2PO_4$, pH 7.4, 287 mM NaCl, 2.7 mM KCl, and 300 mM imidazole), purified by size exclusion chromatography (HiLoad 16/600 Supderdex 75 pg, Cytiva), and stored at –80°C in aliquots (*Figure 1—figure supplement 1*).

### Membrane protein purification

scTMHC2 is a designed single-chain TM homodimer that was previously reported (*Lu et al., 2018*). The SpyTag sequence with a linker GGSGGS at its N- and C-termini was encoded for the DNA handle attachment. The gene block was cloned into pBT7-C-His vector (Bioneer), and the vector was inserted into *E. coli* BL21-Gold(DE3)pLysS by heat shock transformation (40 s at 42°C). The cells were grown in LB broth with 100 µg/ml ampicillin at 37°C until $OD_{600}$ reached to ~1.0. The overexpression was induced by the addition of 200 µM IPTG. The protein was expressed for 18 hr at 18°C, and the cells were harvested by centrifugation (5700 rpm, 10 min, and 4°C). The cell pellets were resuspended in 50 ml of a lysis buffer (25 mM Tris-HCl, pH 7.4, 150 mM NaCl, 10% glycerol, 1 mM TCEP, and 1 mM PMSF). The sample was lysed by the Emulsiflex-C3 at the pressure of 15,000–17,000 psi. 1% of DDM detergent (GoldBio) was added to the lysed cell, and the cells was incubated for 1 hr at 4°C to extract the membrane protein. The cell debris was removed by centrifugation (18,000 rpm, 30 min, and 4°C), and the supernatant was incubated with Ni-IDA resin for 1 hr at 4°C. The resin was packed into a column and was washed with a washing buffer (25 mM Tris-HCl, pH 7.4, 150 mM NaCl, 10% glycerol, 1 mM TCEP, 0.1% DDM, and 20 mM imidazole). The protein was then eluted with an elution buffer (25 mM Tris-HCl, pH 7.4, 150 mM NaCl, 10% glycerol, 1 mM TCEP, 0.1% DDM, and 300 mM imidazole). The purified membrane protein was concentrated to ~10 µM and stored at –80°C in aliquots (*Figure 1—figure supplement 1*).

### DNA handle attachment to membrane proteins

Two types of 1022 bp DNA handles, modified with amine at one end and azide or 2×biotin at the other end, were amplified by PCR from $\lambda$ DNA template (NEB, N3011S). The primers were added to total 8 ml PCR mixture – 1 µM of a forward primer (ACAGAAAGCCGCAGAGCA) with amine at 5' end, 0.5 µM of a reverse primer (TCGCCACCATCATTTCCA) with azide at 5' end, and 0.5 µM of the reverse primer with 2×biotin at 5' end. The DNA handles were purified using HiSpeed Plasmid Maxi kit (Qiagen). For maleimide modification at the amine end, the purified DNA in 1 ml NaHCO$_3$ (pH 8.3) was incubated with 1 mM SM(PEG)$_2$ (Thermo Scientific Pierce) for 20 min at 20–22°C. Unconjugated SM(PEG)$_2$ was removed by Econo-Pac 10DG Column (Bio-Rad). The buffer used for column equilibration and sample elution was 0.1 M sodium phosphate (pH 7.3) with 150 mM NaCl. The DNA handles modified with maleimide were covalently attached to MBP-fused SpyCatcher with a single cysteine (2 hr at 20–22°C for the incubation; ~20 µM for the protein concentration; *Min et al., 2016*). Unconjugated proteins were removed by HiTrap Q HP column (Cytiva) with the gradient mode of 0–1 M NaCl in 20 mM Tris-HCl (pH 7.5). The DNA peak fractions were concentrated to ~350 nM and stored at –80°C in 10 µl aliquots. ~30% of the sample is the SpyCather-conjugated DNA handle (~100 nM). The conjugated DNA handles were attached to the target proteins with SpyTags (1–2 hr at 20–22°C for the incubation; ~70 nM for DNA and ~2 µM for protein). The yield of the final construct with azide at one end and 2×biotin at the other end is ~10% of the SpyCather-DNA construct. The sample was diluted to make ~200 pM DNA handle and stored at –80°C in 10 µl aliquots (*Figure 1—figure supplement 1*). The dilution buffer was 25 mM Tris-HCl, pH 7.4, 150 mM NaCl, 1 mM TCEP, and 1.5% (w/v) total amphiphiles for bicelles composed of DMPC lipids (Avanti) and CHAPSO detergents (Sigma) at a 2.5:1 molar ratio.

### Making DNA hairpin constructs

A 2070-bp DNA handle was amplified by PCR from $\lambda$ DNA template (final 2.5 µg/ml; NEB, N3011S) using a forward primer (TAAGGATGAACAGTTCTGGC) with 2×biotin at 5' end, and a reverse primer (GCAGCGAGTTGTTCC/1',2'-Dideoxyribose/AATGATCCATTAATGGCTTGG) (final 2 µM each). The total 200 µl PCR product was purified with HiGene Gel & PCR Purification System (Biofact, GP104-100) with 50 µl elution buffer (10 mM Tris-HCl, pH 8.0). The purified sample was diluted to 100 nM with deionized water. 50 nM of the DNA handle was mixed with 500 nM of an 80-nt ssDNA forming a hairpin structure (GAACAACTCGCTGCAAACAACAACAAAAGAGTCAACGTCTGGATCTTTTTTGA TCCAGACGTTGACTCAAAAGACATACC). The ssDNA has phosphate at 5' end and azide at 3' end. The nucleic acid sequence at the center (GAGTCAACGTCTGGATCTTTTTTGATCCAGACGTTGACTC) forms the DNA hairpin motif with 17 bp stem and 6T loop. The incubation condition was 1 hr at 37°C in a ligation buffer (50 mM Tris-HCl, pH 7.5, 10 mM MgCl$_2$, 10 mM DTT, and 10 mM ATP). During the incubation, the 5' end linker of the ssDNA (GAACAACTCGCTGCAAACAACAACAAAA) was annealed to the DNA handle. The mixture was further incubated for 16 hr at 16°C with T4 DNA ligase added (Enzynomics, M001S). After the ligation, the ligase was inactivated by incubation at 65°C for 10 min followed by 5 min on ice. The sample was purified with the purification kit with 50 µl of the elution buffer. The ligation yield was ~30%. The ligated product was diluted to 1 nM with 10 mM Tris-HCl (pH 8.0) with 0.1 mM EDTA and stored at –20°C in 10 µl aliquots.

### DBCO modification on magnetic bead surface

The surface of magnetic beads was modified by DBCO using the conjugation between amine and NHS ester. 35 µl of amine-coated magnetic beads (Thermo Fisher Scientific, M-270 amine, 14307D) were washed with buffer A (0.1 M sodium phosphate, pH 7.4, 150 mM NaCl, and 0.01% Tween20) using DynaMag–2 Magnet (Thermo Fisher Scientific, 12321D). After washing, the beads were resuspended with 100 µl of the buffer A. The bead slurry was incubated with 10 µl of 0.25–1 mM DBCO-sulfo-NHS (Merck, 762040; dissolved in DMSO) for 3 hr at 25°C with slow rotation. The bead slurry was washed again and resuspended in 35 µl of the buffer A. The DBCO-modified bead sample was stored at 4°C.

### Surface passivation of sample chamber surfaces

The coverslips of 24×50 mm and 24×40 mm (VWR; No. 1.5) were used for the bottom and top surface of single-molecule sample chamber, respectively. Both coverslips were cleaned by 1 M KOH in sonication bath for 30 min and washed with deionized water. The bottom coverslips (24×50 mm)

were further cleaned by Piranha solution ($H_2SO_4$[98%]:$H_2O_2$[30%]=2:1 volume ratio). The surface of the coverslips was functionalized by amine using the silanization solution of N-(2-aminoethyl)–3-aminopropyltrimethoxysilane (Sigma-Aldrich, 8191720100), acetic acid, and methanol (1:5:100 volume ratio) (*Min et al., 2016*). To this end, the coverslips were incubated with the silanization solution for 30 min at 20–22°C. The amine-functionalized coverslips were washed with methanol, deionized water, and dried with centrifugation (2000 rpm and 5 min). The PEG polymers (Laysan Bio; 1:27.5 molar ratio of Biotin-PEG-SVA-5000 and mPEG-SVA-5000) were conjugated to the surface of the bottom coverslips using the amine-NHS ester conjugation. For this passivation, each 50 µl of PEG solution (total ~40 mM PEG mixture in 100 mM sodium bicarbonate, pH 8.4) was incubated between two coverslips for 4–16 hr at 20–22°C in humidity chambers. The coverslips were washed with deionized water, dried with centrifugation (407 g and 5 min) and stored at –20°C.

## Making single-molecule sample chambers

Single-molecule sample chambers were constructed with the two surface-treated coverslips with double-sided tapes. Its channel volume (1 CV) was ~10 µl. Streptavidin-coated polystyrene beads (Spherotech, SVP-10–5) were washed with buffer B (0.1 M sodium phosphate, pH 7.4, 150 mM NaCl, and 0.1% Tween 20), injected into the chamber, and then incubated for 2–5 min at 20–22°C. The polystyrene beads are attached to the PEG surface via biotin-streptavidin interaction. The beads are used as reference beads for the correction of microscope stage drifts. 100 mg/ml bovine serum albumin was injected into the chamber for further passivation, incubated for 5 min at 20–22°C, and then washed with washing buffer I (for the membrane protein, 50 mM Tris-HCl, pH 7.4, 150 mM NaCl, 0.1% DDM; for the DNA hairpin, 0.1 M sodium phosphate, pH 7.4, 150 mM NaCl). 10 µl sample of ~200 pM target molecule was mixed with 1 µl of 0.04 µM traptavidin and incubated for 15 min at 20–22°C. 1 CV of the sample (final 20–200 pM) was injected into the sample chamber and incubated for 10 min at 20–22°C. To block unoccupied biotin-binding pockets, 1 CV of a 30-nt biotin-labeled oligonucleotide (10 µM in the washing buffer I) was injected into the chamber and incubated for 5 min at 20–22°C. The chamber was then washed with washing buffer II (50 mM Tris-HCl, pH 7.4, 150 mM NaCl, and 0.05% DDM) for the membrane protein or washing buffer I for the DNA hairpin. The DBCO-modified magnetic beads were washed and resuspended with washing buffer II for the membrane protein and washing buffer I for the DNA hairpin (20× diluted). The magnetic beads were injected into the chamber and incubated for 1 hr at 25°C. The chamber was washed with a single-molecule buffer (for the membrane protein, 50 mM Tris-HCl, pH 7.4, 150 mM NaCl, and 2% [w/v] total amphiphiles for bicelles; for the DNA hairpin, 0.1 M sodium phosphate, pH 7.4, and 150 mM NaCl). The bicelles were composed of DMPC lipids (Avanti) and CHAPSO detergents (Sigma) at a 2.5:1 molar ratio.

## Instrumentation of single-molecule magnetic tweezers

Magnetic tweezer apparatuses were built on an inverted microscope (Olympus, IX73) and a motorized XY stage (ASI, MS-2000FT or Prior Scientific, H117P1) as previously described (*Min et al., 2018*; *Min et al., 2015*; *Min et al., 2016*). A light-emitting diode (Thorlabs, M455L4, $\lambda$ =447 nm) was used to illuminate the magnetic beads tethered to target molecules and the reference beads directly attached to the sample chamber surface. Diffraction patterns of the beads were captured using a charge-coupled device camera (JAI, CM-040GE or CM-030GE) at 60–90 Hz. A piezo nanopositioner (Mad City Labs, Nano-F100S) was used to calibrate the bead heights from the surface according to the diffraction patterns by moving the focal plane of an objective lens (Olympus, UPLFLN100XO2 or UPLXAPO100XO) in known increments. The extension of a target molecule represented by the height of a tethered bead was tracked using the $\chi^2$ analysis with the calibration data. The XY stage drifts were corrected from positions of the reference beads immobilized on the chamber surface. Magnetic field gradients were generated by a pair of two permanent neodymium magnets with the antiparallel configuration of magnetic moments (each 10×10×12 mm; separated by 1 mm). The vertical and rotational movements of the magnets were controlled using a translation motor (PI, M-126.PD1 or M-126.PD2) and a rotation motor (PI, DT-50 or DT-34). The mechanical force applied to a bead-tethered molecule was calibrated as a function of the magnet position using the formula $F=k_B T \cdot L/\delta x^2$ derived from the inverted pendulum model (*Sarkar and Rybenkov, 2016*), where $F$ is the applied force, $k_B$ is the Boltzmann constant, $T$ is the absolute temperature, $L$ is the extension, and $\delta x^2$ is the magnitude

of lateral fluctuations (*Figure 4—figure supplement 1*). The imaging rooms of the single-molecule magnetic tweezers were maintained at 20–22°C.

## Polymer model analysis of force-extension curves

The force-extension curves of the membrane protein with DNA handles were median-filtered for the extension (6–9 Hz window) and smoothed for the force (3–4.5 Hz window). The unfolding forces and step sizes were extracted from the force-extension curves and compared with the protein force-extension curves expected for the $U_c$ and $U_h$ states (*Choi et al., 2019*; *Min et al., 2015*). The $U_c$ state was modeled by the Marko–Siggia formula of the WLC model (*Marko and Siggia, 1995*), $FP_c/k_BT = l_c/C_c + (1–l_c/C_c)^{-2}/4 – 1/4$, where $F$ is the applied force, $l_c$ is the protein extension of the $U_c$ state, $P_c$ is the polypeptide persistence length of 0.4 nm (*Choi et al., 2019*; *Min et al., 2015*; *Oesterhelt et al., 2000*), and $C_c$ is the polypeptide contour length that was estimated by the number of unfolded residues of tertiary structure (n=153 for scTMHC2) times the average residue-residue distance of 0.36 nm (*Choi et al., 2019*; *Min et al., 2015*; *Oesterhelt et al., 2000*). For the $U_h$ state, the helices were modeled by the KR formula (*Kessler and Rabin, 2004*), $l_h = –1/2f – \alpha/(f·\tanh(2\alpha))+C_h/\tanh(fC_h)–(2\alpha^2/3f)·(1/\tanh(fC_h)–fC_h/\sinh^2(fC_h)–1)$, $f = F/k_BT$, $\alpha = \sqrt{fC_h^2/4P_h}$, where $l_h$ is the protein extension of the $U_h$ state, $P_h$ is the helix persistence length of 9.17 nm (*Choi et al., 2019*; *Min et al., 2015*), and $C_h$ is the helix contour length that was estimated by the number of helix residues times the average helical rise per residue of 0.16 nm (*Choi et al., 2019*; *Min et al., 2015*). The peptide linkers in the $U_h$ state were modeled by the Marko–Siggia formula. The protein force-extension curves were corrected by the end-to-end distance of the pulling residue points of tertiary structure ($\Delta d$=1.1 nm for scTMHC2) because the extension changes during unfolding are measured smaller by $\Delta d$.

## Extraction of unfolding kinetics from pulling-cycle experiments

The cumulative plots of unfolding force distributions were normalized to the unfolding probabilities as a function of force (*Figure 5—figure supplement 3*). To obtain the values for the unfolding kinetic parameters at zero force ($k_{u0}$ and $\Delta x_f^\dagger$), the unfolding probability profiles were fitted with the formula, $U=1–\exp(\int dF\ [–k_{u0}·\exp(F\Delta x_f^\dagger/k_BT)/\dot{F}])$, derived from the first-order rate equation, $dN/dt = –k_uN$, and the Bell equation, $k_u = k_{u0}·\exp(–F\Delta x_f^\dagger/k_BT)$, where $F$ is the force, $U$ is the unfolded fraction (unfolding probability), $N$ (=1–$U$) is the folded fraction (folding probability), $k_u$ ($k_{u0}$) is the kinetic rate constant at an arbitrary (zero) force, $\Delta x_f^\dagger$ is the distance between the native and transition states, and $\dot{F}$ (=d$F$/d$t$) is the force-loading rate. The d$F$/d$t$ (=$\dot{r}$d$F$/d$z$) is approximately a linear function of the force (a$F$ +b) in the force range of $U$=0–1, where d$F$/d$z$ is the first derivative of force with respect to magnet height that was calibrated by two-term exponential function, and $\dot{r}$ (=d$z$/d$t$) is the magnet speed that was maintained to be constant. The constants of a and b were obtained by fitting the force-loading rate (d$F$/d$t$) as a function of force ($F$). To simplify the formula, $U=1–\exp(\int dF\ [–k_{u0}·\exp(F\Delta x_f^\dagger/k_BT)/(aF +b)])$, the exponential integral function ($\int \exp(x)/x\ dx$) was approximated as the first term of series expansion ($\exp[x]/x$) since higher terms are only ~5% of the first term in the unfolding force ranges. The final equation used to fit the unfolding probabilities as a function of force was derived to $U=1–\exp(–k_{u0}k_BT·\exp[F\Delta x_f^\dagger/k_BT]/[\Delta x_f^\dagger(aF +b)])$. The analysis yields the approximate values of unfolding kinetic parameters of $k_{u0}$ and $\Delta x_f^\dagger$ (*Min et al., 2015*).

## MC simulations for pulling-cycle experiments

The unfolding forces were sampled from the probability density function (PDF) of unfolding as a function of force, i.e., the equation $U=1–\exp(–k_{u0}k_BT·\exp[F\Delta x_f^\dagger/k_BT]/[\Delta x_f^\dagger(aF +b)])$ with set kinetic parameters, which was derived in the previous section. The cumulative force distributions were normalized to unfolding probabilities as a function of force. The unfolding probability profiles were then created in every 50 times of random sampling up to 10,000 times. Each unfolding probability was fitted with the equation to extract the values of unfolding kinetic parameters. The beads tethered to molecules have the error in their diameter (RSD<3%; designated in the manual for the M-270 bead), which causes the error in unfolding forces and resultant probabilities. Thus, two types of MC simulations with or without bead size error were performed as simulation type A and B, respectively (*Figure 5—figure supplement 3*). For the type A, the random sampling was performed for all different PDFs that were experimentally measured, reflecting the bead size error. For the type B, the random sampling was performed only for a median PDF with respect to $F_{U=0.5}$ at different time points, effectively removing

the bead size error. The generated kinetic error plots were averaged to obtain the mean kinetic errors during progress of pulling cycles, as shown in *Figure 5C*.

## Error estimation for unfolding kinetics from pulling-cycle experiments

The error plot of kinetic parameters during progress of pulling cycles shown in *Figure 5D* was obtained from MC simulations for multiple probability profiles with various possible shapes (*Figure 5—figure supplement 5*). All the data were created from the simulation-type B since its results are consistent with those of experiment for the analysis mode 1, as shown in *Figure 5—figure supplement 4*. All the error traces were averaged to generate the expected mean kinetic error plot (black curve in *Figure 5D*). The SE curve represented by the factor of $1/\sqrt{n}$ does not apply in this case because the individual error traces were not sampled from one population (i.e. one unfolding probability profile). The two-term exponential model shows a better fit than the reciprocal square root model, as indicated by the reduced $\chi^2$ used for a goodness-of-fit test (*Figure 5—figure supplement 6*). The regression model adopted in our analysis is a phenomenological model that more properly describes the error decay curve. The trend of the first fast and then slow decay is not unusual because it is also expected for the reciprocal square root model – the plot $1/\sqrt{n}$ decays fast and then slowly, too (*Figure 5—figure supplement 6*).

## Polymer model analysis of the constant-force trace

The 90 Hz extension trace measured for 9 hr at 12 pN was smoothed with 1.8 Hz window for the correction of long-term drift. The long-term drift was removed by subtracting a baseline for the most stable $N$ state regions of the smoothed trace. The original trace with the drift corrected was median-filtered with 4.5 Hz window. The Gaussian peaks of the extension distribution were compared with extension values for the expected structural states estimated by the WLC and KR polymer models. The average penetration angle ($<\theta_p>$) was estimated by $L_{h,exp} = 4D_{helix} \cdot \cos<\theta_p> + \Sigma L_{linker} - \Delta d$, where $L_{h,exp}$ is the measured step size at 12 pN for the $U_h$ state, $D_{helix}$ is the contour length of the helices, $L_{linker}$ is the peptide linker extension at 12 pN obtained from the WLC model, and $\Delta d$ is the end-to-end distance of the pulling residue points of tertiary structure (*Figure 6—figure supplement 3*). The average penetration depth ($<d_p>$) was estimated by $<d_p> = (D_{helix} - <D_{helix,w}>) \cdot \sin<\theta_p>$, where $<D_{helix,w}>$ is the average contour length of the soluble parts of helices bounded by RK ring on the pulling side (*Lu et al., 2018*). The core thickness of DMPC/CHAPSO bicelles was estimated as ~3.0 nm by taking the average of the measured values in previous reports (3.49, 2.45, and 2.88 nm; *Leite et al., 2022*; *Giudice et al., 2022*; *Murugova et al., 2022*). The bicelle core represents the lipid tail part of bicelle.

## Reconstruction of the folding energy landscape

The folding energy landscape at 12 pN was characterized from the Boltzmann relation, $\Delta G(l) = -k_B T \cdot \ln(p[l]/p[l=0])$, where $l$ is the protein extension, $p(l)$ is the probability density of the protein extension, and $\Delta G(l)$ is the free energy change from $l=0$ (*Gebhardt et al., 2010*; *Woodside et al., 2006*). The $p(l)$ was obtained from probability density of the measured extension ($P[l]$) by removing the bead-handle effect using a deconvolution method. We adopted a nonlinear constrained iterative deconvolution method, $p^{(n+1)}(l) = p^{(n)}(l) + r[p^{(n)}(l)] \cdot (P(l) - S(l) * p^{(n)}(l))$, $r[p^{(n)}(l)] = r_0(1 - 2|p^{(n)}(l) - 1/2|)$, where $r_0$ is the amplitude, $S(l)$ is the point spread function (PSF) obtained from the $N$ state at 12 pN, and $n$ is the index of iterations ($r_0=2$ and $n_{max} = 5000$) (*Gebhardt et al., 2010*; *Woodside et al., 2006*). Our method was successfully tested with a two-state extension trace and PSF trace made by MC simulations (*Figure 6—figure supplement 1*). The parameters of simulated traces, such as time constants, noise strength, and total times, mimic those of our measured trace. The dwell times and random noises were extracted from exponential and normal distributions, respectively. The positions of the energy barrier heights ($\Delta G^\dagger$) were located by fitting with the Gaussian function.

## Extraction of (un)folding times

The median-filtered extension trace with 4.5 Hz window was analyzed by the HMM with the number of the Gaussian peaks (*Zhang et al., 2016*; *Jiao et al., 2017*). The observed (un)folding time ($\tau_{obs}$) for each transition was obtained as the dwell time constant for one state until transitioning to another state (*Figure 6—figure supplement 2*). The $\tau_{obs}$ was corrected for the limited temporal resolution using a formula, $\tau_{corr} = \tau_{obs} - t_r$, where $\tau_{corr}$ is the corrected (un)folding time, and $t_r$ is the instrument

time resolution (*Jacobson and Perkins, 2020*). Based on a previous study (*Cossio et al., 2015*), the $\tau_{corr}$ was multiplied by a factor of 0.95 to remove the tethered bead-handle effect and estimate the molecular (un)folding time ($\tau_{(un)folding}$), which is explained as follows. The large μm-sized bead used in typical magnetic tweezers is likely the main source of the error (*Neupane and Woodside, 2016*). The diffusion coefficient for our 2.8 μm bead ($D_q$) is calculated as $1.5\times10^5$ nm²/s, using the Stokes–Einstein equation ($D=k_BT/6\pi\eta R$), where $\eta$ is the dynamic viscosity of the solvent, and $R$ is the radius of a spherical particle (*Hummer and Szabo, 2010*). For this calculation, water viscosity was used for the solvent viscosity since the viscosity of bicelle solutions was observed to remain unchanged within the range of 0.1–12% (w/v) total amphiphiles (*Lu et al., 2012*). The diffusion coefficient for a helical protein (un)folding on the barrier top ($D_x$) was estimated as $10^6$ nm²/s in a previous study (*Yu et al., 2012*). For the approximate lower bound for viscosity ratio of DMPC lipid membranes to water (~$10^2$; *Figure 7—source data 1*), the $D_x$ is reduced to ~$10^4$ nm²/s by the Einstein relation ($D=k_BT/\eta$; *Chung and Eaton, 2013*). The ratio $D_x/D_q$ is then estimated to be ~$10^{-1}$. From the previous theoretical estimates (*Cossio et al., 2015*), the bead-handle effect in this low $D_x/D_q$ ratio regime is approximately by the factor of 0.95.

# Acknowledgements

This work was supported and funded by the National Research Foundation of Korea (NRF-2020R1C1C1003937 to DM) and Ulsan National Institute of Science and Technology (UNIST-1.190147.01 to DM). We thank SY Hong in UNIST for helpful comments and Y Zhang in Yale University for providing the base codes for the HMM analysis.

# Additional information

### Funding

| Funder | Grant reference number | Author |
| --- | --- | --- |
| National Research Foundation of Korea | 2020R1C1C1003937 | Duyoung Min |
| Ulsan National Institute of Science and Technology | 1.190147.01 | Duyoung Min |

The funders had no role in study design, data collection and interpretation, or the decision to submit the work for publication.

### Author contributions

Seoyoon Kim, Resources, Data curation, Software, Formal analysis, Validation, Investigation, Visualization, Methodology, Writing - original draft, Writing - review and editing; Daehyo Lee, Resources, Data curation, Formal analysis, Validation, Investigation, Visualization; WC Bhashini Wijesinghe, Resources, Investigation; Duyoung Min, Conceptualization, Resources, Software, Formal analysis, Supervision, Funding acquisition, Validation, Investigation, Visualization, Methodology, Writing - original draft, Project administration, Writing - review and editing

### Author ORCIDs

Duyoung Min http://orcid.org/0000-0002-2856-8082

### Decision letter and Author response

Decision letter https://doi.org/10.7554/eLife.85882.sa1
Author response https://doi.org/10.7554/eLife.85882.sa2

# Additional files

### Supplementary files
• MDAR checklist

## Data availability

All data and analysis codes that support the findings of this study are available in the manuscript, figure supplements, source data, and source code files.

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
