## [Editor Report]

This work presents an important advance of single-molecule force spectroscopy of membrane proteins providing a new robust design of the linkage between a target single molecule and solid support. The data provide compelling evidence of the improved mechanical stability of the pulling system for more statistically reliable force measurements of bio-macromolecules. Also, the quantification of speed-limit of membrane protein folding is exquisite and informative, representing an important contribution to the field.

---

## [Decision Letter]

**Decision letter after peer review:**

Thank you for submitting your article "Robust single membrane protein tweezers" for consideration by eLife. Your article has been reviewed by 3 peer reviewers, and the evaluation has been overseen by a Reviewing Editor and Kenton Swartz as the Senior Editor. The following individual involved in the review of your submission has agreed to reveal their identity: Heedeok Hong (Reviewer #1).

Essential revisions:

1) Please, provide point-by-point responses to the issues raised by the reviewers and address the issues in the manuscript if necessary.

2) Reframing the uniqueness of the authors' approach by comparing it to the previous single-molecule tweezer studies that employ covalent tethers.

Reviewers 2 and 3 point out that there are multiple reports employing the covalent (including DBCO-azide click chemistry) and traptavidin-biotin x2 tethers to improve the stability of a pulling system (please, see the comments by Reviewer 2 and the references listed by Reviewer 3). I (Reviewing Editor) have an impression that the uniqueness of this work is to combine the covalent tethers with long DNA handles, which achieves a minimal interaction between solid support (including beads) and a target protein as well as a tight connection between them. The authors' approach would be particularly adequate for studying membrane proteins complexed with large amphiphilic assemblies.

I recommend that authors broaden the literature citation including the papers listed by Reviewer 3 and clarify the methodological uniqueness of this work compared to others. Currently, only Popa et al. are cited in this regard, and the papers using the DBCO-azide click chemistry are not cited. This could be achieved by modifying the first and second paragraphs in the Introduction and the contents in the Discussion.

3) Measuring the transition kinetics at a constant force.

This type of experiment is a hallmark of single-molecule force spectroscopic studies but is not presented in this work with the new tethers. Reviewer 3 strongly recommends this experiment as proof of principle. This is a suggestion, but not necessary. If authors decide to do so, I recommend showing a minimal amount of data and analysis. The result would make the paper more compelling.

4) Clarifying technical issues in the result and method sections.

Reviewer 3 brought up many technical questions (the use of a reference bead, the loading rate-force relationship, the calibration of force, the moving rate of beads vs the loading rate, the analysis of error propagation, etc.). I recommend that authors clarify these issues in Result or Methods, if appropriate.

*Reviewer #2 (Recommendations for the authors):*

In this study, the authors have developed methods that allow for repeatedly unfolding and refolding a membrane protein using a magnetic tweezers setup. The goal is to extend the lifespan of the single-molecule construct and gather more data from the same tether under force. This is achieved through the use of a metal-free DBCO-azide click reaction that covalently attaches a DNA handle to a superparamagnetic bead, a traptavdin-dual biotin linkage that provides a strong connection between another DNA handle and the coverslip surface, and SpyTag-SpyCatcher association for covalent connection of the membrane protein to the two DNA handles.

The method may offer a long lifetime for single-molecule linkage; however, it does not represent a significant technological advancement. These reactions are commonly used in the field of single-molecule manipulation studies. The use of multiple tags including biotin and digoxygenin to enhance the connection's mechanical stability has already been explored in previous DNA mechanics studies by multiple research labs. Additionally, conducting single-molecule manipulation experiments on a single DNA or protein tether for an extended period of time (hours or even days) has been documented by several research groups. I recommend that the authors consider submitting their work to a specialized journal that focuses on methodology or experimental protocol.

*Reviewer #3 (Recommendations for the authors):*

– The authors mainly focus on the model membrane protein scTMH2, which is certainly an interesting system. However, they stress that their method also works for much more hydrophilic DNA and mention data for a DNA hairpin. These data, however, seem to not be shown anywhere. It would be nice to add them.

– The main improvement appears to be the more stable and robust tethering approach, compared to previous methods. However, the stability is hard to evaluate from the data provided. The much more common way to test stability in the tweezers is to report lifetimes at constant force(s). Also, there are actually previous methods that report on covalent attachment, even working using DBCO. These papers should be compared. Some relevant references are likely:

Invincible DNA tethers: covalent DNA anchoring for enhanced temporal and force stability in magnetic tweezers experiments.

Janissen R, Berghuis BA, Dulin D, Wink M, van Laar T, Dekker NH. Nucleic Acids Res. 2014 Oct;42(18):e137. doi: 10.1093/nar/gku677.

Copper-free click chemistry for attachment of biomolecules in magnetic tweezers.

Eeftens JM, van der Torre J, Burnham DR, Dekker C. BMC Biophys. 2015 Sep 25;8:9. doi: 10.1186/s13628-015-0023-9.

High-yield, ligation-free assembly of DNA constructs with nucleosome positioning sequence repeats for single-molecule manipulation assays

Yi-Yun Lin, Tine Brouns, Pauline J. Kolbeck, Willem Vanderlinden, Jan Lipfert

bioRxiv 2023.01.05.522917; doi: https://doi.org/10.1101/2023.01.05.522917

– The authors use an attachment to the surface via two biotin-traptavidin linkages. How does the stability of this (double) bond compare to using a single biotin? Engineered streptavidin versions have been studied previously in the magnetic tweezers, again reporting lifetimes under constant force, which appears to be a relevant point of comparison:

Designed anchoring geometries determine the lifetimes of biotin-streptavidin bonds under constant load and enable ultra-stable coupling.

Gruber S , Löf A , Sedlak SM , Benoit M , Gaub HE , Lipfert J . Nanoscale. 2020 Nov 7;12(41):21131-21137. doi: 10.1039/d0nr03665j.

– Very long measurements of protein unfolding and refolding have been reported previously:

A HaloTag Anchored Ruler for Week-Long Studies of Protein Dynamics.

Popa I, Rivas-Pardo JA, Eckels EC, Echelman DJ, Badilla CL, Valle-Orero J, Fernández JM. J Am Chem Soc. 2016 Aug 24;138(33):10546-53. doi: 10.1021/jacs.6b05429. Epub 2016 Aug 9.

and

Multiplexed protein force spectroscopy reveals equilibrium protein folding dynamics and the low-force response of von Willebrand factor.

Löf A, Walker PU, Sedlak SM, Gruber S, Obser T, Brehm MA, Benoit M, Lipfert J. Proc Natl Acad Sci U S A. 2019 Sep 17;116(38):18798-18807. doi: 10.1073/pnas.1901794116. Epub 2019 Aug 28.

Here, too, a comparison would be relevant. In light of this previous work, the statement in the abstract "However, the weak molecular tethers used in the tweezers limit a long time, repetitive mechanical manipulation because of their force-induced bond breakage" seems a little dubious. I do not doubt that there is a need for new and better attachment chemistries, but I think it is important to be clear about what has been done already.

– This is tangential to the current manuscript, but it has actually been demonstrated quite clearly by X-ray scattering that DDM micelles are not spherical (unlike what is often drawn, e.g. here in Figure 2D) but rather oblate ellipsoids. Similarly, bicelles are more mixed and micelle-like than what is drawn in Figure 2.

– Page 5, line 99: If the PEG layer prevents any sticking of beads, how do the authors attach reference beads, which are typically used in magnetic tweezers to subtract drift?

– Figure 3 left me somewhat puzzled. It appears to suggest that the "no detergent/lipid" condition actually works best, since it provides functional "single-molecule conjugation" for two different DBCO concentrations and two different DNA handles, unlike any other condition. But how can you have a membrane protein without any detergent or lipid? This seems hard to believe.

Figure 3 also seems to imply that the bicelle conditions never work. The schematic in Figure 1 is then fairly misleading since it implies that bicelles also work.

– When it comes to investigating the unfolding and refolding of scTMHC2, it would be nice to see some traces also at a constant force. As the authors state themselves: magnetic tweezers have the advantage that they "enable constant low-force measurements" (page 8, line 189). Why not use this advantage?

In particular, I would be curious to see constant force traces in the "helix coil transition zone". Can steps in the unfolding landscape be identified? Are there intermediates?

– I liked the collection of references summarized in Figure 4D and the supplement. However, given that it is well known that the unfolding depends on loading rate, I am not sure how useful this analysis is to make the point that 50 pN is sufficient, in general, "regardless of [...] loading rates" (page 6, line 143). The authors say that the 50 pN upper limit is "arbitrary" (page 6, line 144). The fact is, one could always push for very high loading rates and exceed 50 pN. Note that I do not doubt that the presented attachment scheme is stable enough for a large range of very interesting measurements; in addition, if we want very high loading rates and forces, AFM is preferable anyhow. But I think this claim of generality is misleading.

– Speaking of loading rates and forces: How were the forces calibrated? This seems to not be discussed. And how were constant loading rates achieved? In Figure 4 it is stated that experiments are performed at "different pulling speeds". How is this possible? In AFM (and OT) one controls position and measures force. In MT, however, you set the force and the bead position is not directly controlled, so how is a given pulling speed ensured?

It appears to me that the numbers indicated in Figures 4A and B are actually the speeds at which the magnets are moved. This is not "pulling speed" as it is usually defined in the AFM and OT literature. Even more confusing, moving the magnets at a constant speed, would NOT correspond to a constant loading rate (which seems to be suggested in Figure 4A), given that the relationship between magnet positions and force is non-linear (in fact, it is approximately exponential in the configuration shown schematically in Figure 1).

– Finally, when it comes to the analysis of errors, I am again puzzled. For the M270 beads used in this work, the bead-to-bead variation in force is about 10%. However, it will be constant for a given bead throughout the experiment. I would expect the apparent unfolding force to exhibit fluctuations from cycle to cycle for a given bead (due to its intrinsically stochastic nature), but also some systematic trends in a bead-to-bead comparison since the actual force will be different (by 10% standard deviation) for different beads. Unfortunately, the authors average this effect away, by averaging over beads for each cycle (Figure 4). To me, it makes much more sense to average over the 1000 cycles for each bead and then compare. Not surprisingly, they find a larger error "with bead size error" than without it (Figure 5A). However, this information could likely be used (and the error corrected), if they would only first analyze the beads separately.

What is the physical explanation of the first fast and then slow decay of the error (Figure 5B)? I would have expected the error for a given bead after N pulling cycles to decrease as 1/sqrt(N) since each cycle gives an independent measurement. Has this been tested?

[Editors' note: further revisions were suggested prior to acceptance, as described below.]

Thank you for resubmitting your work entitled "Robust membrane protein tweezers reveal the folding speed limit of helical membrane proteins" for further consideration by eLife. Your revised article has been evaluated by Kenton Swartz (Senior Editor) and a Reviewing Editor. We thank the authors for properly addressing all the concerns raised by reviewers.

In the revision, the newly added experiment, measuring the transition kinetics at constant force as proof of concept, was carried out impressively for 9 continuous hours at 12 pN. Based on the new data, the "speed limit" of membrane protein folding was evaluated, whose slow rate (~two- or three-fold slower than that of globular proteins) was attributed to the high viscosity of the lipid bilayer. This unprecedented finding provides new insights into membrane protein folding and is an important contribution to the field.

The manuscript has been improved but there are some remaining issues that need to be addressed, as outlined below (those are mainly for improving the accessibility to broader audiences):

A) Recommendations for "Abstract":

– Lines 17-19: "However, the weak molecular tethers used for the membrane protein studies have limited long-time, repetitive molecular transitions due to force-induced bond breakage."

Please, emphasize why observing the long-time, repetitive molecular transition is important in single-molecule force spectroscopy. You can modify the structure of the sentence.

– Lines 22-24: "This method is >100 times more stable than a conventional linkage system of 1 digoxigenin (dig) and 1 biotin, allowing for the survival for ~12 h at 50 pN and ~1000 pulling cycle experiments."

Please, clarify the statement, ">100 times more stable". Is the new method more stable regarding the time, applied force, number of pulling cycles, or something else?

Also, I recommend reconsidering the part, "of 1 digoxigene (dig) and 1 biotin", which seems too specific to be added in the Abstract.

– Line 26: and reveal its entire folding pathway, including the hidden dynamics of helix-coil transitions.

Probably, the statement, "its entire folding pathway", needs to be toned down. The word "entire" could be deleted.

– Lines 30: compared to μs-level soluble protein folding.

Please, consider "-scale for soluble protein folding" instead of "-level soluble protein folding".

B) Strengthening "Results: Estimating the speed limits of the folding transitions"

>Please, address the following two questions and change the main text if necessary.

First, it is not clear how the new method and prolonged measurement of folding-unfolding transitions helped to estimate the speed limit. Making a connection between the previous part and this part will make the overall flow more coherent.

Second, related to the point above, readers may ask why the previous single-molecule tweezer studies on membrane protein folding could not evaluate the speed limit.

---

## [Author Response]

Essential revisions:1) Please, provide point-by-point responses to the issues raised by the reviewers and address the issues in the manuscript if necessary.

We have responded to all the reviewers’ comments in proper ways and revised the manuscript accordingly.

2) Reframing the uniqueness of the authors' approach by comparing it to the previous single-molecule tweezer studies that employ covalent tethers.Reviewers 2 and 3 point out that there are multiple reports employing the covalent (including DBCO-azide click chemistry) and traptavidin-biotin x2 tethers to improve the stability of a pulling system (please, see the comments by Reviewer 2 and the references listed by Reviewer 3). I (Reviewing Editor) have an impression that the uniqueness of this work is to combine the covalent tethers with long DNA handles, which achieves a minimal interaction between solid support (including beads) and a target protein as well as a tight connection between them. The authors' approach would be particularly adequate for studying membrane proteins complexed with large amphiphilic assemblies.I recommend that authors broaden the literature citation including the papers listed by Reviewer 3 and clarify the methodological uniqueness of this work compared to others. Currently, only Popa et al. are cited in this regard, and the papers using the DBCO-azide click chemistry are not cited. This could be achieved by modifying the first and second paragraphs in the Introduction and the contents in the Discussion.

We revised Introduction, Results, and Discussion accordingly. The first four paragraphs in the Introduction briefly review previous tweezer methods with an improved stability and delineate where our work is placed. In the first paragraph of the Results, we also briefly discussed how and why our DBCO tethering strategy differs from previous DBCO methods. In the first paragraph of the Discussion, we compared the previous methods regarding the stability improvement.

3) Measuring the transition kinetics at a constant force.This type of experiment is a hallmark of single-molecule force spectroscopic studies but is not presented in this work with the new tethers. Reviewer 3 strongly recommends this experiment as proof of principle. This is a suggestion, but not necessary. If authors decide to do so, I recommend showing a minimal amount of data and analysis. The result would make the paper more compelling.

The manuscript now includes the data from constant force experiments – the structural transitions between four states, energy landscape reconstruction, transition kinetics, and the pre-exponential factor estimation. We extensively edited the main text and Methods accordingly. Relevant figures are Figures 6 and 7, Figure 6–figure supplements 1–3, and Figure 7–source data 1.

4) Clarifying technical issues in the result and method sections.Reviewer 3 brought up many technical questions (the use of a reference bead, the loading rate-force relationship, the calibration of force, the moving rate of beads vs the loading rate, the analysis of error propagation, etc.). I recommend that authors clarify these issues in Result or Methods, if appropriate.

We added an additional section in Methods titled “Instrumentation of single-molecule magnetic tweezers”, which include the force calibration (Figure 4–figure supplement 1A) and how to attach reference beads on the surface. We replaced the terminologies of “loading rate” and “pulling speed” with “magnet speed” that is more appropriate. In Figure 5–figure supplement 6 along with a Method section titled “Error estimation for unfolding kinetics from pulling-cycle experiments”, we responded to the question of the regression analysis. In Figure 5–figure supplement 1, we presented the traces for individual molecules that also show the measurement stability during progress of pulling cycles.

Reviewer #2 (Recommendations for the authors):In this study, the authors have developed methods that allow for repeatedly unfolding and refolding a membrane protein using a magnetic tweezers setup. The goal is to extend the lifespan of the single-molecule construct and gather more data from the same tether under force. This is achieved through the use of a metal-free DBCO-azide click reaction that covalently attaches a DNA handle to a superparamagnetic bead, a traptavdin-dual biotin linkage that provides a strong connection between another DNA handle and the coverslip surface, and SpyTag-SpyCatcher association for covalent connection of the membrane protein to the two DNA handles.The method may offer a long lifetime for single-molecule linkage; however, it does not represent a significant technological advancement. These reactions are commonly used in the field of single-molecule manipulation studies. The use of multiple tags including biotin and digoxygenin to enhance the connection's mechanical stability has already been explored in previous DNA mechanics studies by multiple research labs. Additionally, conducting single-molecule manipulation experiments on a single DNA or protein tether for an extended period of time (hours or even days) has been documented by several research groups. I recommend that the authors consider submitting their work to a specialized journal that focuses on methodology or experimental protocol.

One of the unique features of our work is the development of a robust single-molecule tweezer method that is applicable to membrane proteins, rather than simply making another stable system. As re-written in Introduction, it is not straightforward as we have to consider the membrane reconstitution. We believe that our work is expected to overcome the bottleneck in membrane protein studies that arises when using single-molecule tweezer methods.

To improve the delivery of the contextual information, we revised Introduction, Results, and Discussion. The first four paragraphs in the Introduction briefly review previous tweezer methods with an improved stability and delineate where our work is placed. In the first paragraph of the Results, we also briefly discussed how and why our DBCO tethering strategy differs from previous DBCO methods. In the first paragraph of the Discussion, we compared the previous methods regarding the stability improvement.

Additionally, the revised manuscript now includes new findings – the full dissection of structural transitions of a helical membrane protein, the observation of hidden helix-coil transitions at a constant force, and the estimation of kinetic pre-exponential factors. We believe that the new findings provide important insights into membrane protein folding, in addition to the usefulness of our method itself for membrane protein studies. We extensively edited the main text and Methods accordingly. Relevant figures are Figures 6 and 7, Figure 6–figure supplements 1–3, and Figure 7–source data 1.

Reviewer #3 (Recommendations for the authors):– The authors mainly focus on the model membrane protein scTMH2, which is certainly an interesting system. However, they stress that their method also works for much more hydrophilic DNA and mention data for a DNA hairpin. These data, however, seem to not be shown anywhere. It would be nice to add them.

Those data were already included in the original manuscript. The relevant figures are Figure 4B, Figure 5–figure supplement 2, and Figure 5–figure supplement 3.

– The main improvement appears to be the more stable and robust tethering approach, compared to previous methods. However, the stability is hard to evaluate from the data provided. The much more common way to test stability in the tweezers is to report lifetimes at constant force(s). Also, there are actually previous methods that report on covalent attachment, even working using DBCO. These papers should be compared. Some relevant references are likely:Invincible DNA tethers: covalent DNA anchoring for enhanced temporal and force stability in magnetic tweezers experiments.Janissen R, Berghuis BA, Dulin D, Wink M, van Laar T, Dekker NH. Nucleic Acids Res. 2014 Oct;42(18):e137. doi: 10.1093/nar/gku677.Copper-free click chemistry for attachment of biomolecules in magnetic tweezers.Eeftens JM, van der Torre J, Burnham DR, Dekker C. BMC Biophys. 2015 Sep 25;8:9. doi: 10.1186/s13628-015-0023-9.High-yield, ligation-free assembly of DNA constructs with nucleosome positioning sequence repeats for single-molecule manipulation assaysYi-Yun Lin, Tine Brouns, Pauline J. Kolbeck, Willem Vanderlinden, Jan LipfertbioRxiv 2023.01.05.522917; doi: https://doi.org/10.1101/2023.01.05.522917

As shown in Figure 4E, we evaluated the robustness of our method in a way suggested by you – the lifetime measurement at a constant force. Specifically, ~12 hours at 50 pN. Definitely, our tweezer approach established here is the most robust method for membrane protein studies. Please refer to the section “Assessing robustness of our single-molecule tweezers” in page 7 and line 31.

We discussed the previous covalent methods for which quantitative data are presented in light of the system stability. Please refer to the first paragraph of Discussion. For the case of the second paper, it does not have quantitative data on the stability. For this paper, thus, we briefly mentioned how and why our DBCO tethering strategy differs from that of the paper, in the first paragraph of Results.

– The authors use an attachment to the surface via two biotin-traptavidin linkages. How does the stability of this (double) bond compare to using a single biotin? Engineered streptavidin versions have been studied previously in the magnetic tweezers, again reporting lifetimes under constant force, which appears to be a relevant point of comparison:Designed anchoring geometries determine the lifetimes of biotin-streptavidin bonds under constant load and enable ultra-stable coupling.Gruber S , Löf A , Sedlak SM , Benoit M , Gaub HE , Lipfert J . Nanoscale. 2020 Nov 7;12(41):21131-21137. doi: 10.1039/d0nr03665j.

The paper in this comment showed that the tethering lifetimes of biotin-streptavidin variants were affected by the asymmetric bead anchoring point. However, the situation does not apply to our work as we do not anchor traptavidin to beads. Besides, the stability comparison between the single- and double-biotin systems is not the main point of our work, so we do not have the answer to the question. However, we cited the reference in the first paragraph of Discussion where we discuss the system stability.

– Very long measurements of protein unfolding and refolding have been reported previously:A HaloTag Anchored Ruler for Week-Long Studies of Protein Dynamics.Popa I, Rivas-Pardo JA, Eckels EC, Echelman DJ, Badilla CL, Valle-Orero J, Fernández JM. J Am Chem Soc. 2016 Aug 24;138(33):10546-53. doi: 10.1021/jacs.6b05429. Epub 2016 Aug 9.andMultiplexed protein force spectroscopy reveals equilibrium protein folding dynamics and the low-force response of von Willebrand factor.Löf A, Walker PU, Sedlak SM, Gruber S, Obser T, Brehm MA, Benoit M, Lipfert J. Proc Natl Acad Sci U S A. 2019 Sep 17;116(38):18798-18807. doi: 10.1073/pnas.1901794116. Epub 2019 Aug 28.Here, too, a comparison would be relevant.

We briefly discussed the previous works in the first paragraph of Discussion.

In light of this previous work, the statement in the abstract "However, the weak molecular tethers used in the tweezers limit a long time, repetitive mechanical manipulation because of their force-induced bond breakage" seems a little dubious. I do not doubt that there is a need for new and better attachment chemistries, but I think it is important to be clear about what has been done already.

The sentence is in Abstract, so we also had to consider the conciseness. By simply adding the phrase “used for the membrane protein studies”, we can place our work into a more proper context.

In page 2 and line 3, “…However, the weak molecular tethers used for the membrane protein studies have limited long-time, repetitive molecular transitions due to force-induced bond breakage…”

– This is tangential to the current manuscript, but it has actually been demonstrated quite clearly by X-ray scattering that DDM micelles are not spherical (unlike what is often drawn, e.g. here in Figure 2D) but rather oblate ellipsoids. Similarly, bicelles are more mixed and micelle-like than what is drawn in Figure 2.

We appreciate the comment regarding the micelles, which we have corrected by redrawing them accordingly. Thank you for bringing this to our attention.

We disagree about the bicelles as it is more complex than mentioned. Bicelles vary in size and shape depending on the ratio of lipid to detergent (known as q value), the structure of constituent lipid and detergent, total concentration of amphiphiles, and temperature (ref 1). According to the ref 1, the authors suggested from multiple experiment and simulation methods that the DMPC/DHPC bicelles (6 wt% amphiphile) with a q value below 1.0 deviated from the classic bicelles that contain lipids in the core and detergents in the rim. Thus, the comment is correct if we used the same bicelle condition.

However, we used different constituents and condition for bicelles – the DMPC/CHAPSO bicelles with q = 2.5 and ~1.4 wt% lipid. Morphological studies on DMPC/CHAPSO bicelles are relatively limited, compared to those on DMPC/DHPC bicelles. In a previous work (ref 2), the authors suggested from SANS experiments that the DMPC/CHAPSO bicelles with q = 3.0 and 0.25 wt% lipid would be the classic, bilayer-like bicelles. From a very recent work (ref 3), the authors suggested from SAXS and DLS experiments that the DMPC/CHAPS bicelles with 0.5 wt% lipid are likely the mixed micelles for a q < 1, but the classic bicelles for a q > 1. Additionally, in a previous single-molecule work that our development here bases on (ref 4), the unfolding of a membrane protein under a similar bicelle condition showed a homogeneous population, whereas the unfolding in DDM and CHAPSO micelles showed a highly heterogeneous population (see Supplementary Fig.5 of ref 4). This indicates that the bicelles used in our work are far from the micelles, at least regarding the membrane protein stabilization.

Therefore, although the conditions are not exactly same as that of our work, it is not unreasonable to draw bicelles in the conventional way of the bilayer-like discs for our bicelle condition.

ref (1) Caldwell et al., Low‑q Bicelles Are Mixed Micelles, JPCL, 9, 4469-4473 (2018)

ref (2) Li et al., Morphological Characterization of DMPC/CHAPSO Bicellar Mixtures: A Combined SANS and NMR Study, Langmuir, 29, 15943-15957 (2013)

ref (3) Giudice et al., Expanding the Toolbox for Bicelle-Forming Surfactant–Lipid Mixtures, Molecules, 27, 7628 (2022)

ref (4) Min et al., Mapping the energy landscape for second-stage folding of a single membrane protein, Nature Chemical Biology, 11, 981-987 (2015)

– Page 5, line 99: If the PEG layer prevents any sticking of beads, how do the authors attach reference beads, which are typically used in magnetic tweezers to subtract drift?

The PEG layer consists of biotin-PEG and methyl-PEG at a 1:27.5 molar ratio. As the reference beads are coated with streptavidin, they are attached to the PEG layer by the regular biotin-streptavidin interaction. In page 19 and line 7, you can refer to “…The polystyrene beads are attached to the PEG surface via biotin-streptavidin interaction. The beads are used as reference beads for the correction of microscope stage drifts…”

– Figure 3 left me somewhat puzzled. It appears to suggest that the "no detergent/lipid" condition actually works best, since it provides functional "single-molecule conjugation" for two different DBCO concentrations and two different DNA handles, unlike any other condition. But how can you have a membrane protein without any detergent or lipid? This seems hard to believe.

We explained the raised point in page 6 and line 18,

“…Indeed, the best condition was in the absence of any detergents or lipids (Figure 3; no detergents/lipids only during the conjugation step). This situation is possible because membrane proteins are sparsely tethered to the chamber surface, which kept them from aggregating. However, not using detergents or lipids means that the membrane proteins are definitely deformed from their native folds. Therefore, we sought an optimal solubilization condition for membrane proteins during the DBCO-azide conjugation step...”

Figure 3 also seems to imply that the bicelle conditions never work. The schematic in Figure 1 is then fairly misleading since it implies that bicelles also work.

The buffer conditions shown in Figure 3 are those ONLY during the DBCO-azide conjugation step. In this step, the bicelle conditions did not work. Therefore, after the conjugation in 0.5% DDM, the buffer was exchanged with a bicelle solution. This process is shown in Figure 2 and the finally assembled system is depicted in Figure 1.

To clarify this point, we put a note “Buffer conditions only during the DBCO-azide conjugation step” just above the buffer conditions in Figure 3. You can also find for the relevant exchange step in page 6 and line 31, “…Following a 1 h incubation of the beads in the single-molecule chamber at 25°C, unconjugated beads were washed, and the detergent micelles were exchanged with bicelles to reconstitute the lipid bilayer environment for membrane proteins…”

– When it comes to investigating the unfolding and refolding of scTMHC2, it would be nice to see some traces also at a constant force. As the authors state themselves: magnetic tweezers have the advantage that they "enable constant low-force measurements" (page 8, line 189). Why not use this advantage?In particular, I would be curious to see constant force traces in the "helix coil transition zone". Can steps in the unfolding landscape be identified? Are there intermediates?

Yes, please refer to Figure 6. We were able to dissect three distinct transitions from the fully unstructured state to the native state, including the helix-coil transitions. We also reconstructed the folding energy landscape using a deconvolution method.

Please refer to the pertinent sections in the main text, which are titled “Structural transitions and folding energy landscape over extended time scales” and “Mechanistic dissection of folding transitions”.

– I liked the collection of references summarized in Figure 4D and the supplement. However, given that it is well known that the unfolding depends on loading rate, I am not sure how useful this analysis is to make the point that 50 pN is sufficient, in general, "regardless of [...] loading rates" (page 6, line 143). The authors say that the 50 pN upper limit is "arbitrary" (page 6, line 144). The fact is, one could always push for very high loading rates and exceed 50 pN. Note that I do not doubt that the presented attachment scheme is stable enough for a large range of very interesting measurements; in addition, if we want very high loading rates and forces, AFM is preferable anyhow. But I think this claim of generality is misleading.

We did not intend to argue that “50 pN is sufficient, in general, regardless of loading rates" nor “claim its generality”. To prevent a possible misled interpretation, we slightly edited the sentence as follows (page 7 and line 27). Here, we put the word “used” to make it to be clearly true.

“…We surveyed 54 distinct proteins studied by optical/magnetic tweezers and, for 98% of them, the most probable or average unfolding forces were measured to be less than 50 pN, regardless of their molecular weights and used force-loading rates…”

Just below the sentence, we also put the following sentences to more clarify the point: “Since protein folding involves numerous noncovalent interactions, ranging from hundreds to thousands of inter-residue interactions^63^, most biomolecular interactions could be studied below the 50 pN. Thus, the 50 pN was set as an upper bound for our robustness test…”

63 Gromiha, M. M. and Selvaraj, S. Inter-residue interactions in protein folding and stability. *Prog Biophys Mol Biol*
**86**, 235-277, doi:10.1016/j.pbiomolbio.2003.09.003 (2004).

– Speaking of loading rates and forces: How were the forces calibrated? This seems to not be discussed.

We wrote an additional section in Methods titled “Instrumentation of single-molecule magnetic tweezers”, where we discuss the force calibration. For the actual force calibration data, please see Figure 4–figure supplement 1A.

In page 20 and line 10, “…The mechanical force applied to a bead-tethered molecule was calibrated as a function of the magnet position using the formula *F* = *k*_B_*T∙L*/*δx*^2^ derived from the inverted pendulum model^96^, where *F* is the applied force, *k*_B_ is the Boltzmann constant, *T* is the absolute temperature, *L* is the extension, and *δx*^2^ is the magnitude of lateral fluctuations…”

And how were constant loading rates achieved? In Figure 4 it is stated that experiments are performed at "different pulling speeds". How is this possible? In AFM (and OT) one controls position and measures force. In MT, however, you set the force and the bead position is not directly controlled, so how is a given pulling speed ensured?It appears to me that the numbers indicated in Figures 4A and B are actually the speeds at which the magnets are moved. This is not "pulling speed" as it is usually defined in the AFM and OT literature. Even more confusing, moving the magnets at a constant speed, would NOT correspond to a constant loading rate (which seems to be suggested in Figure 4A), given that the relationship between magnet positions and force is non-linear (in fact, it is approximately exponential in the configuration shown schematically in Figure 1).

You are correct, so we simply modified the “pulling speed” to “magnet speed” in the figure caption. The loading rates provided in the figure (with the notation <F˙>) were average loading rates in 1–50 pN to provide rough estimates. We actually specified it in the caption as “average force-loading rate”. However, this can be misleading at a glance, so we just deleted all the loading-rate values in the figure and caption.

– Finally, when it comes to the analysis of errors, I am again puzzled. For the M270 beads used in this work, the bead-to-bead variation in force is about 10%. However, it will be constant for a given bead throughout the experiment. I would expect the apparent unfolding force to exhibit fluctuations from cycle to cycle for a given bead (due to its intrinsically stochastic nature), but also some systematic trends in a bead-to-bead comparison since the actual force will be different (by 10% standard deviation) for different beads. Unfortunately, the authors average this effect away, by averaging over beads for each cycle (Figure 4). To me, it makes much more sense to average over the 1000 cycles for each bead and then compare. Not surprisingly, they find a larger error "with bead size error" than without it (Figure 5A). However, this information could likely be used (and the error corrected), if they would only first analyze the beads separately.

We might be wrong, but there seems to be a misunderstanding. First, we added Figure 5–figure supplement 1 where you can see individual traces. As expected, the levels of unfolding forces/sizes appear consistent during the progress of pulling cycles. Second, the advantage of averaging for different beads is that you can effectively remove the bead size effect. This “averaging-out” is the key strategy in our kinetic analysis. Based on the error estimation, if you average the values of kinetic parameters obtained from different beads, you can then estimate them with reasonably small errors despite the bead size variations. This becomes more evident after initial hundreds of pulling cycles. The errors for 200 and 1000 cycles are of only ~1% difference, indicating that you do not need to blindly run the pulling cycles. These results are based on the “averaging-out” strategy, which is the merit of our analysis. For more details, please see the section in the main text titled “Assessing statistical reliability of pulling-cycle experiments”, where relevant figures, figure supplements, and Method sections are referred.

What is the physical explanation of the first fast and then slow decay of the error (Figure 5B)? I would have expected the error for a given bead after N pulling cycles to decrease as 1/sqrt(N) since each cycle gives an independent measurement. Has this been tested?

If the sampling was from one population (here, unfolding probability profile), the error would follow a 1/n decay as expected for the standard error. In our analysis, however, we estimated the expected “mean” errors, regardless of detailed shapes of the unfolding probability profiles. To this end, we sampled the data from different possible profiles (shown in Figure 5–figure supplement 5). We then averaged all the error plots to obtain the plot of the mean errors during progress of pulling cycles (black curve in Figure 5D). In this case, the plot does not have to follow the standard error curve represented by the factor 1/n.

We tested this by fitting with the model function of *y* = *A*/n, for various lower limit of *N* = 10, 30, 50, 100, 300, and 500 in the regression analysis (Figure 5–figure supplement 6). The results of the reduced chi-square (χ^2^) used for a goodness-of-fit test (χ^2^ = 1 for the best fit) indicates that the two-term exponential model (χ^2^ = 1.60) shows a better fit than the reciprocal square root model (χ^2^ = 2.30–6.01). The regression model adopted in our analysis is a phenomenological model that more properly describes the error decay curve. The trend of the first fast and then slow decay is not unusual because it is also expected for the reciprocal square root model – the plot 1/n decays fast and then slowly, too (Figure 5–figure supplement 6).

[Editors' note: further revisions were suggested prior to acceptance, as described below.]

In the revision, the newly added experiment, measuring the transition kinetics at constant force as proof of concept, was carried out impressively for 9 continuous hours at 12 pN. Based on the new data, the "speed limit" of membrane protein folding was evaluated, whose slow rate (~two- or three-fold slower than that of globular proteins) was attributed to the high viscosity of the lipid bilayer. This unprecedented finding provides new insights into membrane protein folding and is an important contribution to the field.The manuscript has been improved but there are some remaining issues that need to be addressed, as outlined below (those are mainly for improving the accessibility to broader audiences):A) Recommendations for "Abstract":– Lines 17-19: "However, the weak molecular tethers used for the membrane protein studies have limited long-time, repetitive molecular transitions due to force-induced bond breakage."Please, emphasize why observing the long-time, repetitive molecular transition is important in single-molecule force spectroscopy. You can modify the structure of the sentence.

We slightly edited as follows:

“…However, the weak molecular tethers used for the membrane protein studies have limited the observation of long-time, repetitive molecular transitions due to force-induced bond breakage. The prolonged observation of numerous transitions is critical in reliable characterizations of structural states, kinetics, and energy barrier properties…”

– Lines 22-24: "This method is >100 times more stable than a conventional linkage system of 1 digoxigenin (dig) and 1 biotin, allowing for the survival for ~12 h at 50 pN and ~1000 pulling cycle experiments."Please, clarify the statement, ">100 times more stable". Is the new method more stable regarding the time, applied force, number of pulling cycles, or something else?Also, I recommend reconsidering the part, "of 1 digoxigene (dig) and 1 biotin", which seems too specific to be added in the Abstract.

We replaced the "of 1 digoxigene (dig) and 1 biotin" with “regarding the lifetime”.

– Line 26: and reveal its entire folding pathway, including the hidden dynamics of helix-coil transitions.Probably, the statement, "its entire folding pathway", needs to be toned down. The word "entire" could be deleted.

We deleted the word “entire” accordingly.

– Lines 30: compared to μs-level soluble protein folding.Please, consider "-scale for soluble protein folding" instead of "-level soluble protein folding".

We modified it accordingly.

B) Strengthening "Results: Estimating the speed limits of the folding transitions">Please, address the following two questions and change the main text if necessary.First, it is not clear how the new method and prolonged measurement of folding-unfolding transitions helped to estimate the speed limit. Making a connection between the previous part and this part will make the overall flow more coherent.Second, related to the point above, readers may ask why the previous single-molecule tweezer studies on membrane protein folding could not evaluate the speed limit.

Immediately after submitting, it also occurred to us that adding a paragraph regarding what you mentioned would be helpful in understanding the rationale. Thank you for reminding us. On page 12, line 15, we have added a new paragraph as follows:

“To this end, it is necessary to acquire a large amount of extension data during the transitions using a high temporal resolution. However, the temporal resolution of typical magnetic tweezers is inherently limited due to the camera-based method for the bead position tracking, which restricts the speed limit estimation. Our robust single-molecule method overcomes this limitation by conducting the prolonged observations of numerous transitions, resulting in the collection of abundant data points for each transition event. This experimental strategy would not be feasible with the previous magnetic tweezer method for membrane proteins, due to the less stable molecular tethering under applied forces.”